# KoopSTD: Reliable Similarity Analysis between Dynamical Systems via Approximating Koopman Spectrum with Timescale Decoupling

**Shimin Zhang** [* 1]  **Ziyuan Ye** [* 1]  **Yinsong Yan** [1]  **Zeyang Song** [2 1]  **Yujie Wu** [3]  **Jibin Wu** [1 3 4]

## Abstract

Determining the similarity between dynamical systems remains a long-standing challenge in both machine learning and neuroscience. Recent works based on Koopman operator theory have proven effective in analyzing dynamical similarity by examining discrepancies in the Koopman spectrum. Nevertheless, existing similarity metrics can be severely constrained when systems exhibit complex nonlinear behaviors across multiple temporal scales. In this work, we propose **KoopSTD**, a dynamical similarity measurement framework that precisely characterizes the underlying dynamics by approximating the Koopman spectrum with explicit timescale decoupling and spectral residual control. We show that KoopSTD maintains invariance under several common representation-space transformations, which ensures robust measurements across different coordinate systems. Our extensive experiments on physical and neural systems validate the effectiveness, scalability, and robustness of KoopSTD compared to existing similarity metrics. We also apply KoopSTD to explore two open-ended research questions in neuroscience and large language models, highlighting its potential to facilitate future scientific and engineering discoveries. Code is available at link.

## 1. Introduction

Investigating and comparing non-linear neural networks through the lens of representation space has attracted considerable attention within the machine learning community (Hotelling, 1992; Kriegeskorte et al., 2008; Hardoon et al., 2004; Kornblith et al., 2019; Williams et al., 2021; Klabunde et al., 2025; Huh et al., 2024). Recent developments in representation space analysis have shed light on how internal representations of deep neural networks are influenced by network architectures (Hermann & Lampinen, 2020; Nguyen et al., 2021), training methods (Wang et al., 2018; Mehrer et al., 2020; Csiszárik et al., 2021), and data distributions (Ding et al., 2021; D'Amour et al., 2022). One fundamental assumption behind most existing similarity analyses is that the representation of a network can closely correspond to its functional characteristics. This assumption holds for static network architectures, such as convolutional neural networks (CNNs) and feed-forward networks (FFNs), which do not involve complex internal dynamics.

However, for non-linear dynamical networks such as recurrent neural networks (RNNs), large language models (LLMs), and brain networks, two significant challenges arise concerning this assumption. Firstly, two dynamical systems can exhibit distinct representational structures while preserving computational equivalence in their underlying dynamics (Maheswaranathan et al., 2019; Dubreuil et al., 2022). Secondly, two dynamical systems may demonstrate different dynamical behaviours while maintaining similar representational structures (Russo et al., 2020; Driscoll et al., 2024). To facilitate the comparison of temporal dynamics, various temporal similarity measures have been proposed, such as dynamic time warping (DTW) (Sakoe & Chiba, 1978) and cross-correlation (Rabiner, 1978), which conduct comparisons after addressing temporal misalignments between representations. However, these measures still struggle to tackle the two challenges mentioned above, particularly when the temporal correspondences between sequences cannot be well established.

Recent advances in Koopman operator theory (Koopman, 1931; Kutz et al., 2016; Brunton et al., 2022) offer a promising solution by transforming the analysis of non-linear dynamical systems into a linear, infinite-dimensional space of observables. This transformation enables us to analyze non-linear systems using the extensive array of powerful techniques available in linear systems theory. Several efforts

---

[*]Equal contribution  [1]Department of Data Science and Artificial Intelligence, The Hong Kong Polytechnic University, Hong Kong SAR, China [2]Department of Electrical and Computer Engineering, National University of Singapore, Singapore [3]Department of Computing, The Hong Kong Polytechnic University, Hong Kong SAR, China [4]Research Center of Data Science and Artificial Intelligence, The Hong Kong Polytechnic University, Hong Kong SAR, China. Correspondence to: Jibin Wu <jibin.wu@polyu.edu.hk>.

*Proceedings of the 42nd International Conference on Machine Learning*, Vancouver, Canada. PMLR 267, 2025. Copyright 2025 by the author(s).

have since started to push the boundaries of dynamical similarity metrics via linear operator representations, such as the Koopman spectral kernel-based metric (Fujii et al., 2017), Perron-Frobenius operator-based metric (Ishikawa et al., 2018) and dynamical similarity analysis based on Hankel Alternative View of Koopman (HAVOK-based DSA) (Ostrow et al., 2024). However, existing approaches are not sensitive to cross-scale coupling in temporal dynamics, which limits their ability to capture multi-scale dynamical patterns intrinsic to many physical and biological systems. Moreover, these methods are prone to spectral pollution, where spurious eigenvalues arise in the numerical approximation of the Koopman operator (Lewin & Séré, 2010; Klus et al., 2018; Colbrook et al., 2019; Kostic et al., 2023; Philipp et al., 2023), potentially undermining the reliability of similarity measurements.

To address these limitations, we introduce **KoopSTD**, a framework to measure similarity between dynamical systems by approximating the **Koop**man **S**pectrum of dynamical systems with **T**imescal **D**ecoupling. Specifically, the transition in the original state space is first decomposed into different frequency components using the short-time Fourier transform (STFT) to capture temporal dynamics across multiple timescales. KoopSTD then extracts spatiotemporal patterns by performing Koopman mode decomposition (KMD) on the obtained time-frequency representation, revealing the intricate interplay among decoupled timescales. Finally, the resulting Koopman spectrum is refined by filtering out spurious modes based on the spectral residuals, ensuring a verified and more accurate representation of the underlying dynamics.

Our contributions can be summarized as follows:

- We propose a dynamical similarity metric, KoopSTD, to effectively analyze the similarities between high-dimensional multi-scale nonlinear dynamical systems (detailed in Sec. 3).

- We theoretically demonstrate that KoopSTD remains invariant under a broad class of invertible linear transformations, including isotropic scaling, rotation, and permutation (see Sec.3.4), which ensure the robustness of the proposed metric under common transformations in representation space (Kornblith et al., 2019; Williams et al., 2021; Klabunde et al., 2025).

- We conduct comprehensive experiments on both physical and neural systems to validate the effectiveness (detailed in Sec. 4.2) and robustness (detailed in Sec. 4.3) of KoopSTD, demonstrating significant improvements over existing similarity metrics.

- We apply KoopSTD to analyze fMRI data from the human auditory cortex, revealing a functional hierarchy across five brain regions (detailed in Sec. 4.4). The results show a strong correspondence with the established structural organization of auditory pathways, suggesting a structural-functional coupling in the human auditory cortex.

- We apply KoopSTD to analyze LLMs, specifically GPT-2 and LLaMA, of varying sizes (detailed in Sec. 4.4). Our findings demonstrate that larger-scale models exhibit significantly higher coherence in their internal dynamics compared to smaller variants, providing a fresh perspective of the scaling laws governing LLMs.

The rest of this paper is organized as follows. Section 2 provides a comprehensive review of similarity metrics for both static systems and non-linear dynamic systems. Section 3 introduces our proposed metric and establishes its theoretical groundedness through rigorous analysis. In Section 4, we present our experimental results on both synthetic and real-world datasets, followed by a discussion of our key findings. Section 5 concludes the paper and outlines potential directions for future research.

## 2. Related Work

There are two main research lines in measuring the similarity between dynamical systems: representation-based methods and dynamics-based methods. In the ***representation-based methods***, the linear analysis approaches, particularly canonical correlation analysis (CCA) (Hotelling, 1992), represent one of the earliest attempts. To capture non-linear relationships, one set of approaches draws from information theory, utilizing metrics such as mutual information (Kraskov et al., 2004; Belghazi et al., 2018) and Jensen-Shannon divergence (Lin, 1991). Another line of solutions analyzes geometric relationships in representational spaces by comparing the patterns of neural responses across different conditions, such as representational similarity analysis (RSA) (Kriegeskorte et al., 2008) and centered kernel alignment (CKA) (Kornblith et al., 2019). Although these representation similarity metrics enable cross-modal system comparison, they are sensitive to simple transformations of the representations, such as data perturbations (Ding et al., 2021) or subset shifts (Davari et al., 2023). Similar to (Ding et al., 2021), generalized shape metrics (Williams et al., 2021) provide a distance measurement by defining a rigorous mathematical framework between representations through optimal linear transformations.

However, these representation-based approaches face critical limitations when comparing dynamical systems, even after applying temporal alignment techniques like cross-correlation (Rabiner, 1978) or DTW (Sakoe & Chiba, 1978). These limitations motivate recent advances to shift the com-

parison from static representations to the underlying dynamics of the non-linear systems themselves (*i.e.*, **dynamics-based methods**). Early efforts to compare dynamical systems focused on model-specific approaches. The fixed-point topology analysis was proposed by (Maheswaranathan et al., 2019) to extract universal dynamical properties across RNN architectures via constructing directed graphs from fixed points and their transition probabilities. Similarly, (Chen et al., 2023) introduced inner product-based similarity measures for comparing filter subspace similarity, though this method was limited to systems with identical architectures. To achieve model-agnostic dynamical similarity metrics, researchers developed methods that directly analyze system dynamics. (Fujii et al., 2017) pioneered this direction by proposing Koopman spectral kernels, which generalized Binet-Cauchy kernels to nonlinear dynamical systems through Koopman operator spectral analysis. Similarly, (Ishikawa et al., 2018) constructed a metric using the ratio of empirical Perron-Frobenius operator norms over infinite time, eliminating the need for exponential discounting. More recently, (Ostrow et al., 2024) advanced the field with DSA based on HAVOK, which evaluates similarity by analyzing how vector fields transform under orthogonal maps to identify topologically conjugate systems.

Despite these advances, these model-agnostic operator-theoretic metrics face inherent spectral pollution issues (Lewin & Séré, 2010; Colbrook et al., 2019) when approximating infinite-dimensional operators with finite-dimensional matrices, and operating primarily in the temporal domain, limiting both their practical utility and ability to capture multi-scale dynamics.

## 3. Method

### 3.1. Preliminary on Koopman Operator Theory

The Koopman operator theory provides a powerful framework for representing nonlinear dynamical systems in an infinite-dimensional Hilbert function space. This allows us to leverage powerful linear analysis techniques such as spectral analysis, mode decomposition and linear prediction, while still capturing the full nonlinear dynamics of the original system. For a time-discrete system described by $x[t+1] = \mathcal{F}(x[t])$ that evolves over a state-space $\Omega \subseteq \mathbb{R}^d$ under the mapping $\mathcal{F} : \Omega \to \Omega$, the Koopman operator $\mathcal{K}$ corresponding to observables $g : \Omega \to \mathbb{C}$ is defined as follows:

$$[\mathcal{K}g](x) = (g \circ \mathcal{F})(x), \quad x \in \Omega, \quad g \in L^2(\Omega, \omega). \quad (1)$$

A key aspect of modern Koopman operator theory is Koopman Mode Decomposition (KMD). It breaks apart a spatiotemporal signal into infinite triplets $\{(\lambda_i, \phi_i, v_i)\}_{i=1}^{\infty}$, where $\lambda_i$ represents Koopman eigenvalues that reflects

scalar amplitudes, $\phi_i$ denotes eigenfunctions, and $v_i$ represents Koopman modes,

$$g(x[t]) = \mathcal{K}^t g(x[0]) = \sum_{i=1}^{\infty} \lambda_i^t \phi_i(x[0]) v_i. \quad (2)$$

This decomposition describes the complex flow patterns by a hierarchy of simpler processes, with their linear superposition reconstructing the full system dynamics. The resulting representation allows comparison of any two dynamical systems according to the principle of topological conjugacy (Ostrow et al., 2024; Redman et al., 2024).

While KMD provides a theoretical framework that decomposes a dynamical system into a set of modes associated with eigenvalues of the Koopman operator, its exact computation is often intractable for real-world systems. Dynamic mode decomposition (DMD) (Williams et al., 2015; Schmid, 2022) addresses this limitation by offering a data-driven approach to approximate this decomposition. To approximate the Koopman operator, DMD constructs a finite-dimensional linear matrix $\mathbf{A}$ through least-squares optimization with two snapshots $W_x = \{g(x[0]), g(x[1]), ..., g(x[t-1])\}$ and $W_x' = \{g(x[1]), g(x[2]), ..., g(x[t])\}$ in observable space:

$$\mathcal{K} \approx \mathbf{A} = \arg\min_{\mathbf{A}} \|W_x' - \mathbf{A}W_x\|_F = W_x' W_x^{\dagger}, \quad (3)$$

where $\|\cdot\|_F$ is the Frobenius norm and $\dagger$ denotes the pseudo-inverse. Thus, the spectral components can be obtained by solving $\mathbf{A}\Phi = \Phi\Lambda$, where $\Lambda = \text{diag}(\lambda_1, \lambda_2, ..., \lambda_N)$ is the diagonal matrix containing the Koopman eigenvalues, and $\Phi$ is the corresponding eigenvectors.

### 3.2. KoopSTD Captures Multi-Scale Dynamics in the Eigen-Time-frequency Coordinate

Nonlinear dynamical systems are inherently complex and challenging to analyze directly in the time domain due to their intricate interactions across multiple timescales, or equivalently, their diverse frequency components. As a prominent method in signal processing, the STFT provides a time-frequency representation through windowed Fourier analysis of the time series. Given a time series $\mathbf{X} \in \mathbb{R}^{T \times N_d}$ with sampling time $T$ and feature dimension $N_d$, the STFT can be regarded as a mapping $h$ that produces a time-frequency representation $\mathbf{Z}$,

$$h : \mathbf{X} \to \mathbf{Z}, \quad (4)$$

where $\mathbf{Z} \in \mathbb{R}^{(\frac{T-l}{s}+1) \times N_f}$ with $l$ and $s$ denoting STFT window length and hop size, respectively. $N_f = \left(\frac{l}{2} + 1\right) \cdot N_d$ is the total number of frequency components obtained by concatenating frequency bins across all dimensions. Note that KoopSTD also supports other time-frequency techniques (e.g., wavelet transform).

To effectively capture multi-scale temporal dynamics, Koop-STD leverages Koopman analysis to approximate the evolution of the system within the eigen-time-frequency coordinates defined by the right singular vectors $\mathbf{V}^*$ of $\mathbf{Z}$,

$$\mathbf{Z} = \mathbf{U}\boldsymbol{\Sigma}\mathbf{V}^*, \tag{5}$$

where $*$ denotes the conjugate transpose. Then, we characterize the finite-dimensional operator $\mathbf{A}$ outlined in Eq. (3) based on the temporal snapshots $\mathbf{W}_{\mathbf{V}}, \mathbf{W}'_{\mathbf{V}}$ from singular vectors $\mathbf{V}$, denoted as $\mathbf{A}_{tf}$,

$$\mathbf{A}_{tf} = \mathbf{W}'_{\mathbf{V}}\mathbf{W}_{\mathbf{V}}^{\dagger}. \tag{6}$$

Note that $\mathbf{V}$ forms a basis in the eigen-time-frequency coordinates, extracting dominant temporal modes from the original time-frequency space. Based on these modes, the Koopman operator approximation $\mathbf{A}_{tf}$ in Eq. (6) explicitly quantifies the underlying dynamics across multiple timescales.

### 3.3. KoopSTD Measures the Dynamics Dissimilarity with Reliable Koopmanism

By applying DMD on eigen-time-frequency observations, the approximated operator $\mathbf{A}_{tf}$ captures the evolution on different timescales. To assess the dynamical similarity between two systems, one could employ similarity metrics such as Wasserstein distance or Procrustes analysis on the operators $\mathbf{A}_{tf}$. However, when performing discrete spectral analysis, the occurrence of spurious eigenvalues, which are not associated with the real dynamics of the operator, can lead to spectral pollution. This phenomenon can significantly distort the comparison, weakening the reliability of the similarity measure and potentially leading to inaccurate conclusions about the dynamical behavior of the systems.

Here, we incorporate the advances in spectral error control via Galerkin method (Fletcher & Fletcher, 1984) for linear operators (Williams et al., 2015; Colbrook & Townsend, 2024) to identify and exclude Koopman modes with unreliable spectral residuals. Specifically, given $N_f$ eigenvalue-eigenvector pairs $\{\hat{\lambda}_j, \hat{v}_j\}_{j=1}^{N_f}$ for the approximated Koopman operator $\mathbf{A}_{tf}$, we explicitly measure the accuracy of each candidate eigenpair by computing its squared relative residual as follows:

$$\widehat{\text{res}}(\hat{\lambda}, \hat{v})^2 = \frac{\hat{v}^* \left[ \mathcal{M} - \hat{\lambda}\mathcal{N}^* - \bar{\hat{\lambda}}\mathcal{N} + |\hat{\lambda}|^2\mathcal{O} \right] \hat{v}}{\hat{v}^* \mathcal{N} \hat{v}}, \tag{7}$$

where $\mathcal{M} = \mathbf{W}'^*_{\mathbf{V}}\mathbf{W}'_{\mathbf{V}}$, $\mathcal{N} = \mathbf{W}^*_{\mathbf{V}}\mathbf{W}'_{\mathbf{V}}$, $\mathcal{O} = \mathbf{W}^*_{\mathbf{V}}\mathbf{W}_{\mathbf{V}}$, and $\bar{\lambda}$ denotes the complex conjugate of eigenvalues. A detailed derivation of Eq. (7) is provided in Appendix B.2. This residual quantifies the deviation from the expected spectral properties, thus serving as a standard for accurate mode selection and reliable dynamical similarity metric.

We give a systematic discussion about the Galerkin and alternative methods for estimating the spectral error bound of linear operators in Appendix B.1.

To systematically demonstrate KoopSTD, let us consider two dynamical systems $\mathbf{X}_1[t + 1] = \mathcal{F}_1(\mathbf{X}_1[t])$ and $\mathbf{X}_2[t + 1] = \mathcal{F}_2(\mathbf{X}_2[t])$, where $\mathbf{X}_1 \in \mathbb{R}^{N_{d_1}}$ and $\mathbf{X}_2 \in \mathbb{R}^{N_{d_2}}$, respectively. The data sampled from two systems are first transformed into the eigen-time-frequency coordinate via STFT and SVD, resulting in two embeddings. We subsequently apply Eq. (7) to both embeddings for extracting the top $r$ eigenvalue-eigenvector pairs with the minimum spectral residuals. These pairs represent the evolution of $r$ accurate modes, which capture the system's overall dynamical behaviors. The eigenvalues explicitly reflect the evolving amplitudes of these modes, making it reasonable to use eigenvalues $\{\hat{\lambda}_{1,j}\}_{j=1}^r$ and $\{\hat{\lambda}_{2,j}\}_{j=1}^r$ to measure the discrepancy $d(\mathcal{F}_1, \mathcal{F}_2)$ between two dynamical systems via symmetric permutation invariant distance metrics. Here, we adopt $d(\mathcal{F}_1, \mathcal{F}_2)$ as the Wasserstein distance:

$$d(\mathcal{F}_1, \mathcal{F}_2) = \inf_{\pi \in \mathcal{P}(r)} \left( \frac{1}{r} \sum_{j=1}^{r} \|\hat{\lambda}_{1,j} - \hat{\lambda}_{2,\pi(j)}\|^p \right)^{\frac{1}{p}}, \tag{8}$$

where $\mathcal{P}(r)$ denotes the set consisting of all permutations of $r$ elements. The pseudocode of KoopSTD is summarized in Algorithm 1.

---

**Algorithm 1** KoopSTD Pseudocode

---

**Input:** two time series, $\mathbf{X}_1 \in \mathbb{R}^{T_1 \times N_{d_1}}$ and $\mathbf{X}_2 \in \mathbb{R}^{T_2 \times N_{d_2}}$; STFT window size, $l \in \mathbb{Z}^+$; STFT hop size, $s \in \mathbb{Z}^+$; number of preserved modes, $r \in \mathbb{Z}^+$
**Output:** Dynamics dissimilarity $d$ between $\mathbf{X}_1$ and $\mathbf{X}_2$
**Procedure** $\text{DMD}_{STFT}(\mathbf{X}, l, s)$
    $\mathbf{Z} = \text{STFT}(\mathbf{X}, l, s)$
    Solve $\mathbf{Z} = \mathbf{U}\boldsymbol{\Sigma}\mathbf{V}^*$
    Approximate $\mathbf{A}_{tf}$ by Eq. (6)
    **Return** $\mathbf{A}_{tf}$
**End Procedure**
**Procedure** $\text{RESCONTROL}(\mathbf{A}_{tf}, r)$
    Solve $\mathbf{A}_{tf}\Phi = \Phi\Lambda$ for eigenpairs $\{\hat{\lambda}_j, \hat{v}_j\}_{j=1}^{N_f}$
    **for** $j = 1$ to $N_f$ **do**
        Compute the residual of $\{\hat{\lambda}_j, \hat{v}_j\}$ by Eq. (7)
    **end for**
    Top $r$ accurate eigenvalues $\Lambda = \text{diag}(\hat{\lambda}_1, \hat{\lambda}_2, \ldots, \hat{\lambda}_r)$
    **Return** $\Lambda$
**End Procedure**
$\mathbf{A}_{\mathbf{tf,1}} \leftarrow \text{DMD}_{STFT}(\mathbf{X}_1, l, s)$
$\mathbf{A}_{\mathbf{tf,2}} \leftarrow \text{DMD}_{STFT}(\mathbf{X}_2, l, s)$
$\Lambda_1 \leftarrow \text{RESCONTROL}(\mathbf{A}_{tf,1}, r)$
$\Lambda_2 \leftarrow \text{RESCONTROL}(\mathbf{A}_{tf,2}, r)$
Compute the dynamics dissimilarity $d$ by Eq. (8)

---

### 3.4. Transformation-Invariant Property

In this section, we prove that KoopSTD exhibits fundamental invariance properties, ensuring its robustness against common transformations in the representation space. To facilitate a more concise and unified theoretical analysis, we reformulate the KoopSTD in Eq. (8) into a more general form as shown in Definition 1.

**Definition 1.** (KoopSTD). *Let $\mathcal{F}_1, \mathcal{F}_2 : \mathbb{R}^{N_d} \to \mathbb{R}^{N_d}$ be two dynamical systems with their finite-dimensional approximations of Koopman operators $\mathbf{A}_1, \mathbf{A}_2$. The dynamics dissimilarity between two systems can be defined as:*

$$d(\mathcal{F}_1, \mathcal{F}_2) \triangleq d(\lambda(\mathbf{A}_1), \lambda(\mathbf{A}_2)), \tag{9}$$

*where $\lambda(\cdot)$ denotes the eigenvalue spectrum and $d(\cdot, \cdot)$ can be symmetric permutation invariant metrics (e.g., Wasserstein distance and Jensen–Shannon divergence).*

Building upon the KoopSTD formalized in Definition 1, a natural follow-up question is whether this similarity metric possesses any theoretical groundedness? Our answer, in Proposition 1, is yes: the similarity measurement of KoopSTD remains invariant under invertible linear transformations on the representational space.

**Proposition 1.** *Let $\mathbf{X}_1[t + 1] = \mathcal{F}_1(\mathbf{X}_1[t])$ and $\mathbf{X}_2[t + 1] = \mathcal{F}_2(\mathbf{X}_2[t])$ be two time-discrete dynamical systems, governed by mappings $\mathcal{F}_1, \mathcal{F}_2 : \mathbb{R}^{N_d} \to \mathbb{R}^{N_d}$. Then, the dissimilarity $d(\mathcal{F}_1, \mathcal{F}_2)$ measured by KoopSTD is invariant under a broad class of invertible linear transformations $\mathcal{T}$ on the representational space, such that:*

$$d(\mathcal{T}(\mathcal{F}_1, \mathcal{F}_2)) = d(\mathcal{F}_1, \mathcal{F}_2). \tag{10}$$

Note that $\mathcal{T}$ includes, but is not limited to, some common transformations, such as isotropic scaling, rotations, and permutations (Kornblith et al., 2019; Williams et al., 2021; Klabunde et al., 2025), which frequently appear in real-world neural data or network representation spaces. As a result, KoopSTD demonstrates several notable properties:

**Property 1.** *(Isotropic scaling invariance). KoopSTD remains invariant for any isotropic scaling $\mathcal{T}_{IS}$ on the representation space,*

$$d(\mathcal{T}_{IS}(\mathcal{F}_1, \mathcal{F}_2)) = d(\mathcal{F}_1, \mathcal{F}_2), \tag{11}$$

*where $\mathcal{T}_{IS} = \{\mathbf{X} \mapsto \mathbf{X}\mathbf{Q} : \mathbf{Q} = q\mathbf{I}_{N_d}\}$ and $q \in \mathbb{R}_+$ denotes the scaling factor.*

**Property 2.** *(Rotation invariance). KoopSTD remains invariant for any rotation transformations $\mathcal{T}_R$ on the representation space,*

$$d(\mathcal{T}_R(\mathcal{F}_1, \mathcal{F}_2)) = d(\mathcal{F}_1, \mathcal{F}_2), \tag{12}$$

*where $\mathcal{T}_R = \{\mathbf{X} \mapsto \mathbf{X}\mathbf{Q} : \mathbf{Q} \in O(N_d)\}$, and $O(N_d) := \{\mathbb{R}^{N_d \times N_d}, \mathbf{Q}^T\mathbf{Q} = \mathbf{I}_{N_d}\}$ denote the orthogonal group.*

**Property 3.** *(Permutation invariance). KoopSTD remains invariant for any permutation transformations $\mathcal{T}_P$ on the representation space,*

$$d(\mathcal{T}_P(\mathcal{F}_1, \mathcal{F}_2)) = d(\mathcal{F}_1, \mathcal{F}_2), \tag{13}$$

*where $\mathcal{T}_P = \{\mathbf{X} \mapsto \mathbf{X}\mathbf{Q}_\pi : \pi \in P_{N_d}\}$, $P_{N_d}$ denote the set of all permutations on $\{1, ..., N_d\}$, and for $\pi \in P_{N_d}$, $\mathbf{Q}_\pi \in \mathbb{R}^{N_d \times N_d}$ denote the permutation matrix.*

The proof of Proposition 1 and Property 1-3 are provided in Appendix A.

## 4. Experimental Results

In this section, we first introduce the experimental setup in Sec. 4.1. We then conduct a comprehensive comparison between the proposed KoopSTD and existing metrics in Sec. 4.2, utilizing classical Lorenz systems, followed by an ablation study of KoopSTD. To verify the robustness of KoopSTD, we further conduct experiments under two challenging scenarios in Sec. 4.3: **a)** different system dynamics yield similar sampled representations, or **b)** similar system dynamics manifested in different sampling representations. Finally, in Sec. 4.4, we apply KoopSTD to address open-ended research questions in neuroscience and LLMs.

### 4.1. Experimental Setup

To evaluate the performance of KoopSTD, we compare it with four existing similarity metrics, including two representation-based methods, CKA (Kornblith et al., 2019) and Procrustes analysis (Williams et al., 2021), as well as two dynamics-based methods, Cross-Correlation (CC) (Rabiner, 1978) and HAVOK-based DSA (Ostrow et al., 2024). Details on their implementation can be found in Appendix C.1. Each metric maps the dynamical patterns into a low-dimensional metric space through pairwise dissimilarity measures, where an effective metric would result in samples of the same dynamical class forming well-separated clusters. The quality of these clusters, and thus the effectiveness of the metrics, is evaluated using the *Silhouette coefficient*. For a given similarity metric $\mathfrak{M}$ applied to a $N$-samples dataset with $k$ distinct dynamical behavior clusters, the coefficient can be expressed as:

$$\mathcal{I}(\mathfrak{M}) = \frac{1}{N} \sum_{i=1}^{N} \frac{b(i) - a(i)}{max(a(i), b(i))}, \tag{14}$$

where $a(i)$ is the average distance between sample $i$ and other samples in the same cluster (intra-cluster distance), and $b(i)$ is the minimum average distance between sample

*Figure 1.* Comparison of different metrics across five classes of Lorenz systems via MDS projection. The more separable the patterns are, the better the performance of the metric.

$i$ and all samples in the nearest neighboring cluster (inter-cluster distance). The Silhouette coefficient ranges from -1 to 1, where values close to 1 indicate well-defined clusters, values close to 0 suggest overlapping clusters, and values close to -1 imply misclassification.

| Metrics / Systems | Representational | | Dynamical | | |
|---|---|---|---|---|---|
| | CKA | Procrustes | CC | HAVOK | KoopSTD |
| Lorenz Systems | -0.05 | -0.04 | -0.27 | 0.47 | **0.94** |
| PDM Attractors | -0.04 | -0.02 | -0.30 | 0.90 | **0.99** |
| Flip-Flop RNNs | 0.20 | 0.98 | -0.16 | 0.10 | **-0.04** |

*Table 1.* Quantitative analysis of existing similarity metrics across three systems using the *Silhouette coefficient*. For the experiments of Lorenz system and PDM attractors, higher values indicate better performance, while in the 3-bit Flip-Flop RNNs experiments, values approach to 0 are preferred. A detailed description of these metrics is provided in Appendix C.1.

### 4.2. Effectiveness Analysis

The Lorenz system is defined by three coupled nonlinear differential equations: $\frac{dx}{dt} = \sigma(y - x)$, $\frac{dy}{dt} = x(\rho - z) - y$, and $\frac{dz}{dt} = xy - \beta z$. By setting $\sigma = 10$, $\beta = \frac{8}{3}$, and varying $\rho$, we obtain five classes of trajectories: line-like ($\rho = 10$), ring-like ($\rho = 20$), two periodic orbits ($\rho = 152, 220$), and chaotic ($\rho = 75$). These trajectories exhibit behaviors ranging from stable to highly nonlinear with multi-scale dynamics, making them ideal for testing various similarity metrics.

As shown in Table 1, KoopSTD achieves the highest Silhouette coefficient (0.94) among all compared similarity metrics, indicating its superiority in analyzing multi-scale dynamics. Figure 1 illustrates that KoopSTD, when visualized through multidimensional scaling (MDS) (Kruskal, 1964), maintains low intra-cluster distances and high inter-cluster distances with the lowest variance. Contrastingly, representation-based metrics (*i.e.*, CKA and Procrustes analysis) fail to differentiate between distinct dynamics. The two dynamics-based metrics (*i.e.*, CC and HAVOK-based DSA) can separate line-like and ring-like dynamics but struggle to differentiate between periodic and chaotic dynamics due to

their shared recurrent patterns in broader temporal scales.

To further understand the contribution of each component in KoopSTD, we conduct *ablation studies*, which are detailed in Appendix E. The results confirm that both timescale decoupling and spectral residual control in KoopSTD are crucial for distinguishing different dynamical behaviors. Specifically, the time-frequency transformation enables effective separation between chaotic dynamics and other types by decoupling complex temporal patterns across multiple scales. Moreover, the spectral residual control for Koopman operator rank reduction improves the distinction between globally similar but locally different dynamics, particularly for periodic orbits, by ensuring the selection of accurate and relevant dynamical modes.

### 4.3. Robustness Analysis

In this section, we compare KoopSTD with other metrics under challenging conditions where either **a)** different system dynamics yield similar sampled representations, or **b)** similar system dynamics manifested in different sampling representations.

**Geometric Homogeneity.** The first experiment aims to assess whether the proposed metric can accurately capture the underlying dynamical disparity despite relatively similar trajectories. Following (Ostrow et al., 2024), we examine three noisy attractors: bistable attractors (BA), line attractors (LA), and point attractors (PA), representing unstable, stable, and leaky integration in perceptual decision-making, respectively. Through adversarial optimization (Galgali et al., 2023), the condition-averaged trajectories of LA and PA become almost identical to BA's, leaving only noise-induced deviations during recurrence to reveal their distinct dynamics. In this experiment, we generated 50 systems for each attractor class with 100 sampled trajectories per system.

According to the intra- and inter-cluster distances shown in Figure 2, Procrustes analysis identifies all samples to be the same due to its emphasis on geometric aspects, where all samples exhibit similar trajectories in the 2D space. While HAVOK-based DSA and KoopSTD are both capable of separating different dynamical patterns, KoopSTD exhibits stronger consistency on the inter-cluster distance across dif-

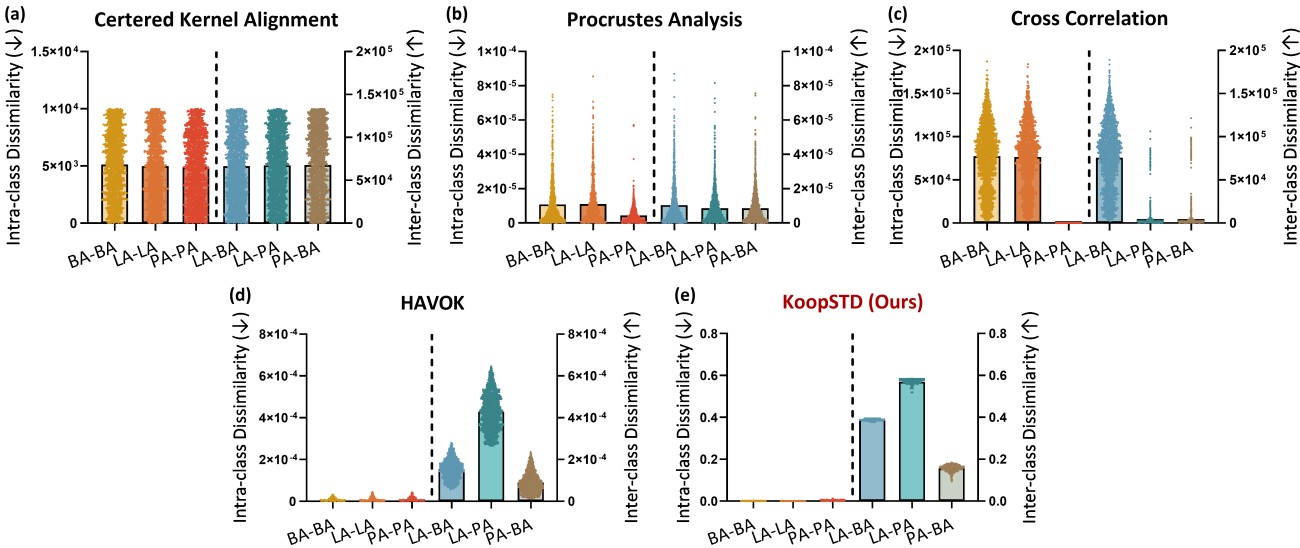

*Figure 2.* Comparison of different metrics between attractor systems. Each class (BA: bistable attractor, LA: line attractor, PA: point attractor) contains 50 systems with 100 sampled trajectories per system. The bars show both intra-class distances within each attractor type and inter-class distances between attractor types. Better metrics are characterized by *lower intra-class distances* (left part of each subfigure) and *higher inter-class distances* (right part of each subfigure).

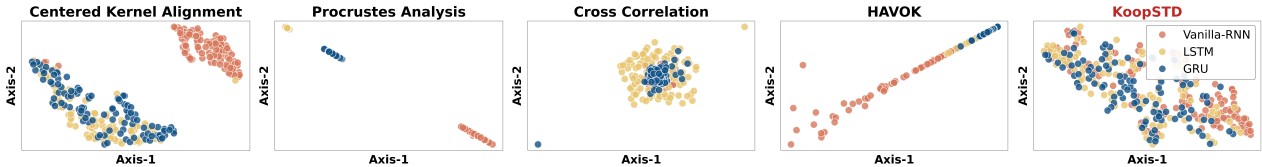

*Figure 3.* Comparison of metrics across three types of RNN via MDS projection. The more inseparable the patterns are, the better.

ferent samples. As a result, KoopSTD achieves an almost perfect clustering performance with a Silhouette coefficient of 0.99. In contrast to KoopSTD, representation-based metrics fail to effectively distinguish both inter-class and intra-class differences in neural dynamics.

**Geometric Disparity.** The second experiment investigates whether the metric can correctly identify neural network systems with the same recurrent dynamic despite differences in geometric network architectures. To explore this, we trained 60 one-layer RNNs with varying architectures on the widely recognized 3-bit Flip-Flop task (Sussillo & Barak, 2013). More details for the task and training can be found in Appendix D. Notably, Figure 10 shows that when projecting network hidden states onto their first three principal components, trajectories across different RNN architectures consistently trace the vertices of a three-dimensional cube, revealing a shared computational structure.

As shown in Figure 3, all existing metrics identify the disparity between networks in solving the 3-bit Flip-Flop task to varying degrees, with CKA, CC, and HAVOK-based DSA detecting partial differences, and Procrustes Analysis detecting the full extent. This points to their reduced robustness

in capturing dynamic similarity amidst geometric variations. On the other hand, KoopSTD yields a Silhouette coefficient of -0.04, indicating minimal separation in their underlying dynamics, further demonstrating its capacity to effectively uncover the shared intrinsic dynamics across different network architectures.

### 4.4. Empower Scientific Discoveries through KoopSTD

KoopSTD is a powerful tool that can facilitate complex dynamical system analysis across various domains. We next demonstrate its potential for scientific research in both computational neuroscience and LLMs by answering two fundamental questions: **RQ1**. To what extent can the KoopSTD reveal functional correspondences among distinct regions of the human auditory cortex? **RQ2**. What novel perspectives can KoopSTD offer regarding the relationship between model size and the functional behavior of LLMs?

**Discover Structural-functional Relationship between Cortical Regions**

We investigated the functional dissimilarity patterns across human auditory cortical regions using KoopSTD on the

"Narratives" dataset (Nastase et al., 2021). This dataset comprises cortical fMRI recordings of 68 individuals listening to the same audio story. Experimental details are provided in Appendix C.5.

As illustrated in Figure 4, our analysis reveals a distinct hierarchical organization of functional relationships among different auditory areas, including primary core area A1, three surrounding belt regions (*i.e.*, LBelt, MBelt, and PBelt) and RI in the anterior insular cortex. Specifically, the diagonal block of A1, LBelt, and MBelt regions exhibit high functional similarity, suggesting shared computational properties in early auditory processing stages. In contrast, the PBelt and RI regions show substantially higher functional dissimilarity compared to other auditory areas, implying their distinct functional role in the auditory processing hierarchy. We leave more comparisons in Appendix D.

Intriguingly, these functional dissimilarities analyzed by KoopSTD correspond closely to the known myelination patterns in the human auditory cortex (Glasser et al., 2016; Baker et al., 2018), where regions exhibit similar myelin concentrations also display similar functional properties (see Appendix Figure 11). This alignment between structural and functional organization provides fresh insights into the relationship between cortical myelination and functional specialization in auditory processing.

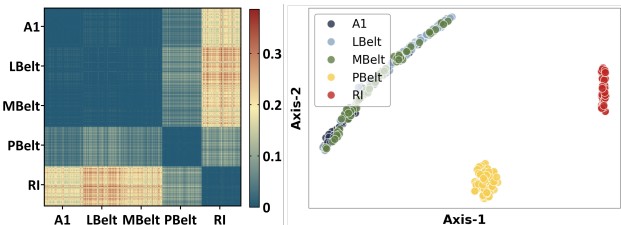

*Figure 4.* Functional dissimilarity between different auditory regions (A1, LBelt, MBelt, PBelt, and RI) measured by KoopSTD.

**Validating LLM Scaling Laws and Beyond**

To make an effective similarity comparison, we randomly selected 75 instructions from the Databricks Dolly 15K dataset (Conover et al., 2023). For each instruction, we conducted experiments using four GPT-2 (Radford et al.,

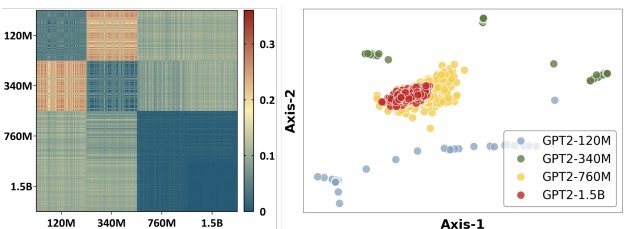

*Figure 5.* Dynamics dissimilarity between different scales (120M, 340M, 760M, and 1.5B) of **GPT-2** measured by KoopSTD.

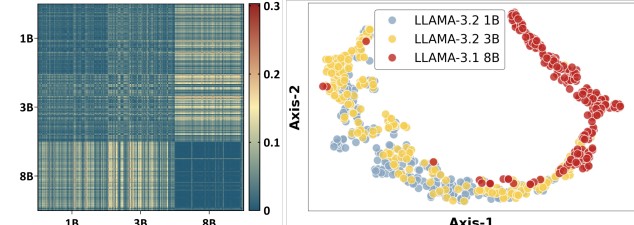

*Figure 6.* Dynamics dissimilarity between different scales (1B, 3B, and 8B) of **LLaMA** measured by KoopSTD.

2019) models with increasing sizes (120M, 340M, 760M, 1.5B parameters). We performed 5 independent inference runs with different random seeds for each instruction, collecting 375 samples from the last layer of each model (75 instructions × 5 runs), resulting in a total of 1500 samples.

As shown in Figure 5, KoopSTD effectively captures significant dynamical discrepancies between different scales of GPT-2 models. Notably, the dynamics of representations produced by larger models (*i.e.*, GPT-2 760M and 1.5B) show higher consistency across different instructions. This is reflected in both the KoopSTD-based dissimilarity matrix and the low-dimensional visualization, where larger models exhibit more concentrated clusters, suggesting potentially more stable and consistent dynamical patterns in their representation spaces. Moreover, in the low-dimensional space, representations from larger models exhibit substantial overlap. These observations demonstrate a hypothesis: '*Representation dynamics evolve towards greater consistency and stability as the model scale increases.*'

To validate this hypothesis beyond GPT-2, we conducted additional experiments on the LLaMA family of models (Dubey et al., 2024) with 1B, 3B, and 8B parameters using 50 instructions randomly sampled from the Massive Multitask Language Understanding (MMLU) dataset (Hendrycks et al., 2021). The similarity results of LLaMA in Figure 6 share the same phenomenon as GPT-2 to some extent, as shown by the convergent dynamics in representation space when scaling up model parameters. These findings shed light on the nuanced differences between LLMs of varying sizes, offering new insights into the scaling laws of LLM (Kaplan et al., 2020).

## 5. Conclusion

In this work, we introduce KoopSTD, a similarity measurement framework for analyzing dynamical similarity between nonlinear systems across multiple timescales via Koopman operator theory. Through extensive experiments on physical systems, neural systems, brain networks, and language models, we demonstrate the effectiveness and broad applicability of KoopSTD. Looking forward, KoopSTD shows promise as a quantitative metric for knowledge distillation in large

language models, potentially guiding the development of more efficient architectures while preserving dynamic behaviors. The framework could be broadly applied to analyze complex dynamical systems across disciplines, from characterizing conformational changes in molecular dynamics and protein folding to investigating neural dynamics in different cognitive states and pathological conditions. These applications may provide valuable insights into the fundamental principles governing both biological and artificial dynamical systems.

## Acknowledgements

This work was supported by the National Natural Science Foundation of China (Grant No. 62306259), Research Grants Council of the Hong Kong SAR (Grant No. C5052-23G, PolyU25216423, and PolyU15217424), and The Hong Kong Polytechnic University (Project IDs: P0043563, P0046094)

## Impact Statement

The proposed KoopSTD framework represents a significant advance in analyzing complex dynamical systems, with implications that extend beyond its technical contributions. This tool has the potential to accelerate scientific discoveries across multiple disciplines by enabling more accurate analysis of complex systems in neuroscience, physics, and artificial intelligence. In neuroscience, KoopSTD's ability to analyze brain dynamics could assist in the early detection of neurological disorders through the identification of atypical dynamical patterns, potentially improving clinical outcomes. Within AI research, the framework could guide the development of more efficient neural architectures by providing deeper insights into internal dynamics, potentially reducing computational costs and the environmental impact of AI systems.

However, these benefits come with important considerations that warrant careful attention. The computational complexity of KoopSTD may limit its accessibility to researchers with limited computational resources, potentially exacerbating existing inequalities in research capabilities. While the framework provides quantitative measures of dynamical similarity, interpreting these results requires significant domain expertise, raising the risk of misuse or overconfidence in conclusions if not properly understood. Additionally, there is a potential for the tool to be misapplied in drawing premature conclusions about complex systems without appropriate statistical rigour. While acknowledging these limitations, we believe the potential benefits of KoopSTD outweigh the risks when used responsibly, particularly in advancing our understanding of complex dynamical systems and developing more efficient AI technologies. The framework's success will ultimately depend on the research community's commitment to addressing these challenges through continued development and responsible application.

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

# A. Proofs of Theoretical Properties

To give better proof to the theoretical analysis, we move the similarity metric provided in the main manuscript here.

*Restatement of Definition* 1. (KoopSTD). Let $\mathcal{F}_1, \mathcal{F}_2 : \mathbb{R}^{N_d} \to \mathbb{R}^{N_d}$ be two dynamical systems with their finite-dimensional approximations of Koopman operators $\mathbf{A}_1, \mathbf{A}_2$. The dynamics dissimilarity between the two systems is defined as:

$$d(\mathcal{F}_1, \mathcal{F}_2) \triangleq d\left(\lambda(\mathbf{A}_1), \lambda(\mathbf{A}_2)\right), \tag{15}$$

where $\lambda(\cdot)$ denotes the eigenvalue spectrum and $d(\cdot, \cdot)$ can be any symmetric, permutation-invariant metric (e.g., Wasserstein distance or Jensen–Shannon divergence).

## A.1. Notations

*Table 2.* Summary of Notations

| Symbol | Description | Symbol | Description |
|--------|-------------|--------|-------------|
| $\mathcal{K}$ | Koopman operator | $\mathcal{F}$ | Original nonlinear mapping functions |
| $\omega$ | probability measure of $L^2 \, \mathcal{H}$ | $\Omega$ | Original state-space |
| $g$ | Observable function | $\circ$ | Composition function |
| $(\lambda, \phi, \mathbf{v})$ | KMD triplet | $\mathbf{A}$ | Approximated operator |
| $W, W'$ | Snapshot data | $\Lambda, \Phi$ | Eigenvalue and eigenvector of $\mathbf{A}$ |
| $X$ | Sampled data in original space | $\mathbf{Z}$ | Time-frequency embedding |
| $N_d$ | The original feature dimension | $T$ | The time dimension |
| $N_f$ | The frequency dimension | $\mathbf{A}_{tf}$ | Approximated operator by KoopSTF |
| $\{\hat{\lambda}_j, \hat{v}_j\}_{j=1}^{N_f}$ | Eigenvalue-eigenvector pair of $\mathbf{A}_{tf}$ | $\mathcal{M}, \mathcal{N}, \mathcal{O}$ | See Eq. (7) |
| $B$ | Size of the dataset | $d(\mathcal{F}_1, \mathcal{F}_2)$ | The dissimilarity between two systems |
| $p$ | Power parameter in Wasserstein distance | $r$ | The number of preserved eigenpairs |
| $l$ | STFT window length | $s$ | STFT hop length |

## A.2. Proof to the Transformation-Invariant Property

Let $\mathbf{X}_1[t + 1] = \mathcal{F}_1(\mathbf{X}_1[t])$ and $\mathbf{X}_2[t + 1] = \mathcal{F}_2(\mathbf{X}_2[t])$ be two time-discrete dynamical systems with state variables $\mathbf{X}_1, \mathbf{X}_2 \in \mathbb{R}^{N_d}$, and they are governed by mappings $\mathcal{F}_1, \mathcal{F}_2 : \mathbb{R}^{N_d} \to \mathbb{R}^{N_d}$. Now we prove that the distance $d(\mathcal{F}_1, \mathcal{F}_2)$ between two systems calculated by KoopSTD remains invariant under invertible linear transformations $\mathcal{T}$, such that:

$$d(\mathcal{T}(\mathcal{F}_1, \mathcal{F}_2)) = d(\mathcal{F}_1, \mathcal{F}_2), \tag{16}$$

where $\mathcal{T} = \{\mathbf{X} \mapsto \mathbf{XQ} : \mathbf{Q} \in GL(N_d, \mathbb{R})\}$. $GL(N_d, \mathbb{R})$ denotes the general linear group of all invertible matrices $\mathbf{Q} \in \mathbb{R}^{N_d \times N_d}$.

*Proof.* Consider a dynamical system governed by the discrete-time evolution equation $\mathbf{X}[t + 1] = \mathcal{F}(\mathbf{X}[t])$, where $\mathbf{X}[t] \in \mathbb{R}^{N_d}$ represents the state of the system at time $t$. The KoopSTD seeks to approximate the Koopman operator by firstly mapping $\mathbf{X} \in \mathbb{R}^{T \times N_d}$ into a time-frequency representation $\mathbf{Z} \in \mathbb{R}^{(\frac{T-l}{s}+1) \times (\frac{l}{2}+1) \times N_d}$. Specifically, the entry $\mathbf{Z}_{m,k,n}$ corresponding to time frame $m$, frequency index $k$ and feature index $n$ is given by:

$$\mathbf{Z}_{m,k,n} = \sum_{t=0}^{l-1} \mathbf{X}[t, n] w[t - m] cos(2\pi kt/l), \tag{17}$$

where $l, s$ respectively denote the length and step size of window function $w[\cdot]$ in STFT. By the linearity of the STFT, this transformation can be expressed in matrix form as

$$\mathbf{Z} = \text{STFT}(\mathbf{X}) = \mathcal{L}_{tf} \mathbf{X}, \tag{18}$$

where $\mathcal{L}_{tf}$ is the transformation matrix representing the STFT operation. Next, KoopSTD computes the right singular vectors $\mathbf{V}$ of $\mathbf{Z}$ by solving SVD:

$$\mathbf{Z}^*\mathbf{Z} = (\mathbf{V}\boldsymbol{\Sigma}\mathbf{U}^*)(\mathbf{U}\boldsymbol{\Sigma}\mathbf{V}^*) = \mathbf{V}\boldsymbol{\Sigma}^2\mathbf{V}^*. \tag{19}$$

The Koopman operator $\mathbf{A}$ can then be approximated based on temporal snapshots (*i.e.*, $\mathbf{W_V}$ and $\mathbf{W_V}'$) of the right singular vectors $\mathbf{V}$ of $\mathbf{Z}$:

$$\mathbf{A} = \mathbf{W}_\mathbf{V}'\mathbf{W}_\mathbf{V}^\dagger. \tag{20}$$

For the transformed representation $\mathbf{XQ}$, we can also derive its time-frequency embedding $\mathbf{Z}_\mathcal{T}$ by substituting $\mathbf{X} = \mathbf{XQ}$ into Eq. (18):

$$\mathbf{Z}_\mathcal{T} = \text{STFT}(\mathbf{XQ}) = \mathcal{L}_{tf}\mathbf{XQ} = \mathbf{ZQ}. \tag{21}$$

Similarly, to obtain $\mathbf{V}_\mathcal{T}$, we substitute $\mathbf{Z} = \mathbf{Z}_\mathcal{T}$ into Eq. (19):

$$\mathbf{Z}_\mathcal{T}^*\mathbf{Z}_\mathcal{T} = (\mathbf{ZQ})^*\mathbf{ZQ} = \mathbf{Q}^*\mathbf{Z}^*\mathbf{ZQ} = (\mathbf{Q}^*\mathbf{V})\boldsymbol{\Sigma}^2(\mathbf{V}^*\mathbf{Q}). \tag{22}$$

Then, we derive the transformed right singular vectors $\mathbf{V}_\mathcal{T}$ and the temporal snapshots $\mathbf{W}_{\mathbf{V},\mathcal{T},}, \mathbf{W}_{\mathbf{V},\mathcal{T}}'$ as:

$$\mathbf{V}_\mathcal{T} = \mathbf{Q}^*\mathbf{V}, \mathbf{W}_{\mathbf{V},\mathcal{T}} = \mathbf{Q}^*\mathbf{W_V}, \mathbf{W}_{\mathbf{V},\mathcal{T}}' = \mathbf{Q}^*\mathbf{W}_\mathbf{V}'. \tag{23}$$

Finally, we express the transformed approximated Koopman operator in terms of $\mathbf{A}_\mathcal{T}$:

$$\mathbf{A}_\mathcal{T} = \mathbf{W}_{\mathbf{V},\mathcal{T}}'\mathbf{W}_{\mathbf{V},\mathcal{T}}^\dagger = \mathbf{Q}^*\mathbf{AQ}. \tag{24}$$

We observe that applying invertible linear transformations to the representational space within the KoopSTD framework corresponds to a **similarity transformation** of the approximated Koopman operator. Consequently, the transformed operator $\mathbf{A}_\mathcal{T}$ is similar to the original $\mathbf{A}$, implying that they share the same eigenvalues:

$$\lambda(\mathbf{A}_\mathcal{T}) = \lambda(\mathbf{A}). \tag{25}$$

According to Definition 1, we have the following equivalence:

$$d(\mathcal{T}(\mathcal{F}_1, \mathcal{F}_2)) = d(\lambda(\mathbf{A}_{1,\mathcal{T}}), \lambda(\mathbf{A}_{2,\mathcal{T}})) = d(\lambda(\mathbf{A}_1), \lambda(\mathbf{A}_2)) = d(\mathcal{F}_1, \mathcal{F}_2). \tag{26}$$

Thus, we have proven that KoopSTD is invariant under a broad class of invertible linear transformations.

$\square$

Since the following transformations on the feature dimension are also invertible linear transformations, we can deduce that KoopSTD remains invariant to the following transformations,

- **Isotropic scaling**: $\mathcal{T}_{IS} = \{\mathbf{X} \to \mathbf{XQ} : \mathbf{Q} = qI_{N_d}, q \in \mathbb{R}^+\}$.

- **Rotations**: $\mathcal{T}_R = \{\mathbf{X} \to \mathbf{XQ} : \mathbf{Q} \in O(N_d)\}$, where $O(N_d) := \{\mathbb{R}^{N_d \times N_d}, \mathbf{Q}^T\mathbf{Q} = I_{N_d}\}$ denotes the orthogonal group.

- **Permutations**: $\mathcal{T}_P = \{\mathbf{X} \to \mathbf{XQ}_\pi : \pi \in P(N_d)\}$, where $P(N_d)$ is the set of permutations on $\{1, \ldots, N_d\}$.

## B. Koopman Spectral Residual Control

### B.1. Related Works for Spectral Error Estimation

Spectral error estimation is critical for evaluating the fidelity of Koopman operator approximations, as spectral pollution can distort interpretations of system dynamics. Such artifacts commonly result from finite-dimensional truncation and suboptimal basis selection in approximation schemes (Klus et al., 2018; 2020; Philipp et al., 2023).

In KoopSTD, we adopt the Galerkin approximation (Williams et al., 2015), a projection-based technique that has demonstrated effectiveness in Residual Dynamic Mode Decomposition (ResDMD) (Colbrook & Townsend, 2024), to suppress spectral pollution and eliminate spurious modes in the Koopman spectrum. By restricting the approximation to a well-chosen subspace, this method enables finer control over spectral content and yields more robust representations of the system's dynamics. Nonetheless, the performance of Galerkin methods is inherently constrained by the expressiveness and scalability of the basis functions (Davies & Stephens, 1983). A limited basis can potentially fail to capture intricate dynamics, while increasing the basis size significantly raises computational costs, undermining efficiency.

Beyond Galerkin methods, several alternative approaches have been proposed to estimate spectral error bounds of Koopman operators. For instance, sparse operator learning frameworks (Hou et al., 2023) target mixing stochastic processes and leverage sparsity in reproducing kernel Hilbert spaces (RKHS) to balance interpretability and sample efficiency. On the other hand, kernel-based methods (Klus et al., 2018; 2020; Kostic et al., 2022; Philipp et al., 2023) offer rigorous convergence guarantees for the spectral approximation of Koopman or transfer operators, and are particularly suited for a variety of stochastic settings, including Markov chains and stochastic differential equations. These methods offer complementary trade-offs in terms of approximation accuracy, theoretical guarantees, computational scalability, and applicability to stochastic dynamics. In future work, we plan to investigate the integration of these techniques into the KoopSTD framework to enhance its robustness and precision in quantifying dynamical similarity.

### B.2. Detailed Derivation of Eq. (7)

Here, we provide a step-by-step derivation of Eq. (7). ResDMD (Colbrook & Townsend, 2024) aims to discard modes associated with large spectral residuals, as determined by their inconsistency with the spectral convergence properties predicted by DMD's quadrature-based approximation. Specifically, for a candidate eigenvalue-eigenvector pair $(\hat{\lambda}, \hat{v})$ of $\mathcal{K}$, the corresponding eigenfunction $\hat{\phi}(x) = \sum_{i=1}^{N_f} g_i(x)\hat{v}_i$, where $N_f$ represents the number of observables $g(\cdot)$, the accuracy of $(\hat{\lambda}, \hat{v})$ can be estimated by the squared residual:

$$
\begin{aligned}
\text{res}(\hat{\lambda}, \hat{v})^2 &= \int_\Omega |\mathcal{K}\hat{\phi}(x) - \lambda\hat{\phi}(x)|^2 \, d\omega(x) \\
&= \langle (\mathcal{K} - \lambda)\hat{\phi}, (\mathcal{K} - \lambda)\hat{\phi} \rangle_\omega \\
&= \langle \mathcal{K}\hat{\phi}, \mathcal{K}\hat{\phi} \rangle_\omega - \langle \hat{\lambda}\hat{\phi}, \mathcal{K}\hat{\phi} \rangle_\omega - \langle \mathcal{K}\hat{\phi}, \hat{\lambda}\hat{\phi} \rangle_\omega + \langle \hat{\lambda}\hat{\phi}, \hat{\lambda}\hat{\phi} \rangle_\omega \\
&= \langle \mathcal{K}g\hat{v}, \mathcal{K}g\hat{v} \rangle_\omega - \hat{\lambda}\langle g\hat{v}, \mathcal{K}g\hat{v} \rangle_\omega - \hat{\lambda}\langle \mathcal{K}g\hat{v}, g\hat{v} \rangle_\omega + |\hat{\lambda}|^2 \langle g\hat{v}, g\hat{v} \rangle_\omega.
\end{aligned}
\tag{27}
$$

In the framework of KoopSTD, the terms $\mathcal{K}g$ and $g$ respectively denote snapshots $W_V'$ and $W_V$. Then, by substituting $\mathcal{M} = W_V'^* W_V'$, $\mathcal{N} = W_V^* W_V'$ and $\mathcal{O} = W_V^* W_V$, we can derive:

$$
\begin{aligned}
\text{res}(\hat{\lambda}, \hat{v})^2 &= \langle \mathcal{K}g\hat{v}, \mathcal{K}g\hat{v} \rangle_\omega - \hat{\lambda}\langle g\hat{v}, \mathcal{K}g\hat{v} \rangle_\omega - \hat{\lambda}\langle \mathcal{K}g\hat{v}, g\hat{v} \rangle_\omega + |\hat{\lambda}|^2 (g\hat{v}, g\hat{v})_\omega \\
&= \hat{v}^*[W_V'^* W_V' - \hat{\lambda} W_V^* W_V' - \bar{\hat{\lambda}} W_V'^* W_V + |\hat{\lambda}|^2 W_V^* W_V]\hat{v} \\
&= \hat{v}^*[\mathcal{M} - \hat{\lambda}\mathcal{N}^* - \bar{\hat{\lambda}}\mathcal{N} + |\hat{\lambda}|^2\mathcal{O}]\hat{v}.
\end{aligned}
\tag{28}
$$

Finally, the squared relative residual can be expressed as:

$$
\widehat{\text{res}}(\hat{\lambda}, \hat{v})^2 = \frac{\|\mathcal{K}\hat{\phi} - \hat{\lambda}\hat{\phi}\|^2}{\|\mathcal{K}\hat{\phi}\|^2} = \frac{\hat{v}^*[\mathcal{M} - \hat{\lambda}\mathcal{N}^* - \bar{\hat{\lambda}}\mathcal{N} + |\hat{\lambda}|^2\mathcal{O}]\hat{v}}{\hat{v}^*\mathcal{N}\hat{v}}.
\tag{29}
$$

## C. Detailed Experimental Settings

### C.1. Alternative Similarity Metrics for Comparison

To demonstrate the versatility of the proposed KoopSTD in measuring dynamical similarity, we compare its clustering performance against four existing similarity metrics based on their calculated dissimilarity matrices from datasets. Below, we provide details for each metric:

- **Dynamical similarity analysis based on Hankel Alternative View of Koopman (HAVOK-based DSA)** Although research on Koopman-based similarity analysis began a decade ago, HAVOK-based DSA proposed by (Ostrow et al.,

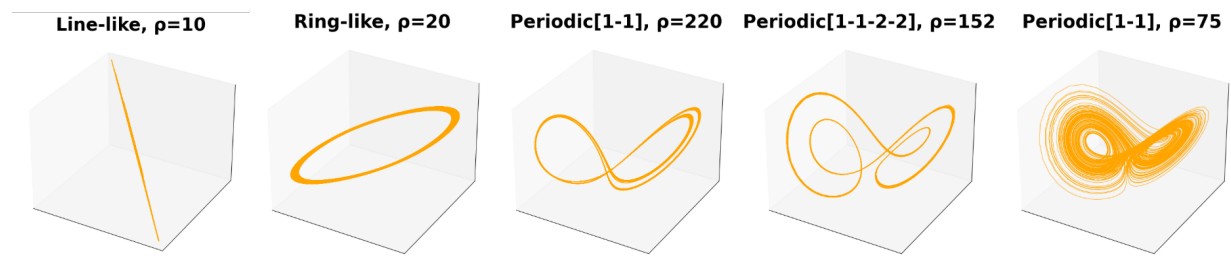

*Figure 7.* Our dataset contains five distinct dynamical behaviors generated by the Lorenz system, each represented by its corresponding trajectory in the phase space. The "Periodic [1-1-2-2]" trajectory refers to a periodic orbit where the first equivalent point is visited twice, followed by the second equivalent point being visited twice. In Figure 1, Periodic [1-1] and Periodic [1-1-2-2] are respectively denoted by Periodic-1 and Periodic-2.

2024) forwards a step in leveraging the theoretically powerful HAVOK framework (Brunton et al., 2017) to capture the intrinsic dynamics of measured systems. To achieve a linear representation of the dynamics, HAVOK first constructs a Hankel representation by applying time-delay embedding to the original data. It then fits truncated-rank regression models on the singular vectors of the Hankel matrix. The chosen truncation rank is claimed to provide an optimal linear fit, forming a Koopman-invariant subspace. For more details of HAVOK, please refer to (Brunton et al., 2017). For the implementation of HAVOK-based DSA, we utilize the official GitHub repository provided by (Ostrow et al., 2024). The results presented in our paper reflect the best performance achieved after careful hyperparameter tuning, including the number of delays, delay interval, and truncation rank.

- **Cross-correlation (CC)** Cross-correlation is defined as the normalized dot product between time series and it can be efficiently computed using *numpy.correlate*. In our study, we define the dissimilarity between two dynamics as the reciprocal of the maximum correlation value.

- **Procrustes Analysis** Procrustes Analysis is a specialized form of generalized shape metrics (Williams et al., 2021), which quantifies the dissimilarity between two representations by optimally aligning them through orthogonal transformation. By minimizing the sum of squared differences between corresponding elements, Procrustes Analysis serves as a powerful tool for comparing geometric structures while preserving their relative spatial relationships. The optimization in our study is implemented by Adam optimizer with a learning rate of 0.01.

- **Centered Kernel Alignment (CKA)** CKA quantifies the similarity between representations by evaluating the alignment of their kernel matrices after centering. Since empirical evidence suggests that using a nonlinear kernel in CKA does not offer significant advantages over the linear kernel (Davari et al., 2023), we opt for the linear kernel in our study to maintain simplicity and efficiency.

### C.2. Lorenz System

We use the Lorenz system to generate a dataset consisting of time series with various dynamic behaviors in the phase space. Specifically, we set $\sigma = 10$ and $\beta = \frac{8}{3}$, while varying $\rho$ to produce five distinct types of trajectories: line-like ($\rho = 10$), ring-like ($\rho = 20$), periodic orbit [1-1] ($\rho = 220$), periodic orbit [1-1-2-2] ($\rho = 152$), and chaotic ($\rho = 75$).

For each $\rho$, we run simulations starting from the same initial point $(-8, 8, 27)$ for a total of 800 seconds, with a time step of $dt = 1e - 3$. We then retain the last 300 seconds, assuming that the dynamic behavior has stabilized. Their trajectories are visualized in Figure 7. Next, we construct a dataset by extracting observations from these trajectories. Specifically, we randomly clip 20-second segments and generate 30 time series instances, resulting in a total of 150 samples. Each sample represents the system's evolution over 20,000 timesteps in $\mathbb{R}^3$.

We apply KoopSTD to first transform the data by STFT with a window length of 500 and a hop length of 1. Next, we fit the DMD model and preserve only the 10 eigenpairs of the approximated Koopman operator with the minimum spectral residuals for the subsequent pairwise dissimilarity computation.

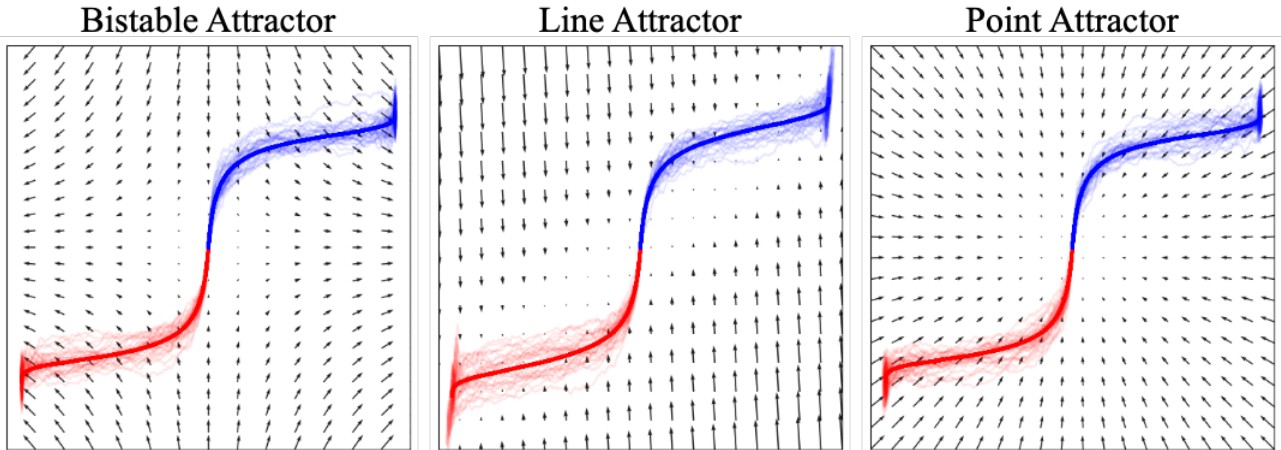

*Figure 8.* The trajectories of single-trial samples (faint) and their condition-averaged counterparts (bold) for three Perceptual Decision-making attractors (reproduced from (Ostrow et al., 2024)). The background black arrows indicate the vector field of each attractor system.

## C.3. Perceptual Decision-making Attractors

We visualize the single-trial samples' trajectories on the 2D space alongside their condition-average, red and blue respectively denote different processes for making different decisions. While the condition-averaged trajectories of the three attractor systems are optimized to appear nearly indistinguishable (detailed implementation refer to (Galgali et al., 2023; Ostrow et al., 2024)), their underlying dynamics are markedly different, as highlighted by the vector fields (background arrows). Specifically, the vector field of a bistable attractor has two stable fixed points (representing two decisions), with trajectories converging toward one of these points based on initial conditions. In contrast, a line attractor's vector field directs the dynamics along a specific trajectory, with no tendency to diverge. A point attractor, on the other hand, has a vector field that points inward toward a single fixed point, indicating that all trajectories globally converge to this point.

The dataset comprises 150 attractor systems, each with 100 trials, where each trial consists of two conditions evolving for 10,000 timesteps in $\mathbb{R}^2$. The proposed KoopSTD effectively separates three distinct attractor systems using the STFT with a window size of 1,024, and a hop length of 128. Additionally, we retain the top 5 eigenpairs from the fitted operator matrix obtained via DMD.

## C.4. 3-bit Flip-Flop Recurrent Neural Networks

In the 3-bit Flip-Flop task, as shown by Figure 9, three separate input streams (grey) deliver occasional pulses with values of -1 or 1 at random intervals. The network is required to retain the most recent pulse value from each channel and continuously output it until a new pulse arrives (dark green solid line), triggering a state switch. The task tests the network's ability to maintain memory over time while adapting to unpredictable and asynchronous changes, pushing its capacity for dynamic memory retention and updating.

We train a population of RNNs with three network architectures: a vanilla RNN with tanh activation, an LSTM, and a GRU. Each network consists of a single recurrent layer with 256 hidden units, followed by a linear readout layer. Following the experimental setup in (Schuessler et al., 2024), we inject zero-mean, isotropic white noise into the hidden states during training to promote robust and stable recurrent dynamics. The initial recurrent and output weights are drawn from centered normal distributions. Each network is trained with a batch size of 32 and a learning rate of 0.01, with different random seeds. To ensure task performance, we only save checkpoints where the MSE falls below 0.001.

According to (Maheswaranathan et al., 2019), while the geometry of RNN activations can be highly sensitive to variations in network architectures, the underlying computational dynamics—such as the topological structure of fixed points—often remain consistent. To explore this, we visualize the hidden states of well-trained RNNs with different architectures in Figure 10. By reducing the dimensionality to three using PCA, we observe that the resulting trajectories exhibit qualitatively similar structures across architectures, tracing the corners of a three-dimensional cube. This cube is further characterized by eight stable fixed points (Golub & Sussillo, 2018), highlighted in black.

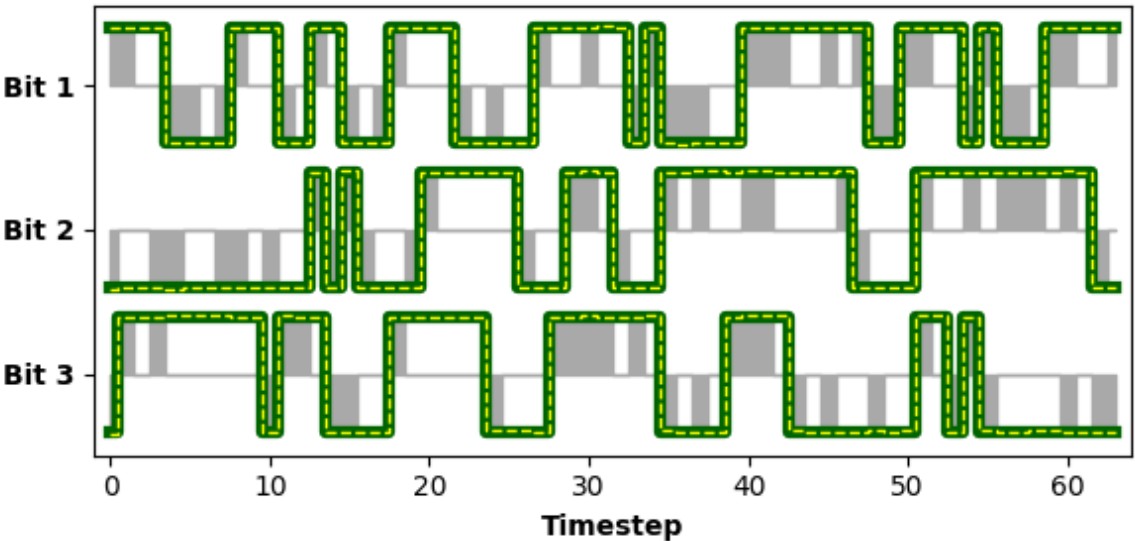

*Figure 9.* Illustration of the input-output scheme for the 3-bit Flip-Flop task. The input pulse sequence is shown in dark gray, the ground truth is represented by a dark green solid line, and the output of a well-trained model is depicted by a yellow dashed line.

**Vanilla RNN**       **LSTM**       **GRU**

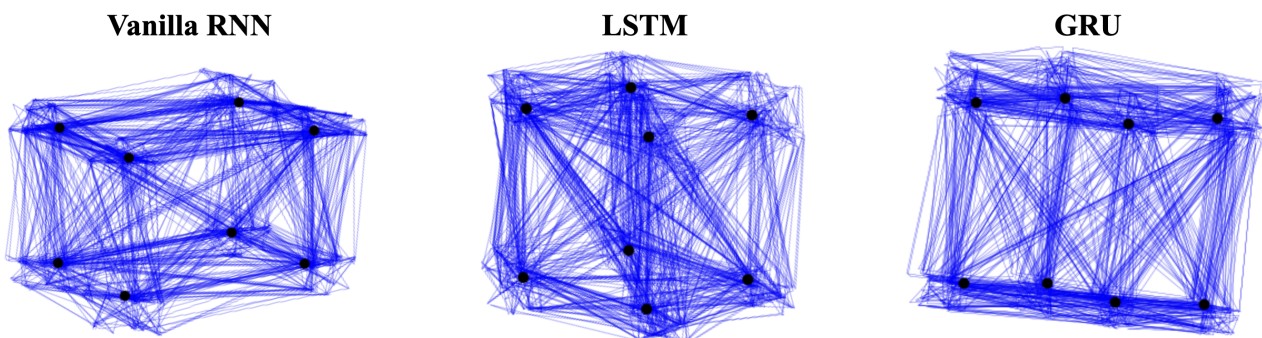

*Figure 10.* The PC3 hidden states trajectories of Vanilla RNN, LSTM, and GRU when solving 3-bit Flip-Flop task.

For each trained network, we extract hidden layer outputs using the same batch of inputs. The hidden representation for each trial is denoted as $h \in \mathbb{R}^{125 \times 256}$, where 125 corresponds to the time dimension. We present the result of KoopSTD with parameters $l = 10, s = 1, r = 10$; however, our method remains robust to hyperparameter selection.

### C.5. Auditory Cortex

Our experiments utilize the Narratives dataset (Nastase et al., 2021), a comprehensive collection of auditory story-listening fMRI data comprising recordings from 345 unique participants across 28 naturalistic spoken stories, each lasting approximately 3 to 56 minutes. For details on data acquisition and preprocessing please refer to the original paper. To analyze auditory cortical regions, we extract fMRI recordings from five areas—A1, LBelt, MBelt, PBelt, and RI—defined on the cortical surface in fsaverage6 space using a multimodal cortical parcellation. For simplicity, we use a subset of the data, specifically from 68 participants listening to the "Pieman" story, which has a duration of 450 seconds. With an fMRI sampling rate of 0.67 Hz, each recording consists of 300 time points.

With $l = 10, s = 1, r = 50$, the result illustrated in Figure 4 by KoopSTD can be reproduced.

### C.6. Large Language Models

**GPT-2.** GPT-2 (Generative Pre-trained Transformer 2) (Radford et al., 2019) is a large-scale language model developed by OpenAI, designed for natural language understanding and text generation. It is based on the Transformer architecture and trained on a diverse corpus of internet text using unsupervised learning. GPT-2 features multiple model sizes, ranging from 124M to 1.5B parameters, and generates coherent, contextually relevant text given a prompt. Unlike traditional task-specific models, GPT-2 excels in zero-shot and few-shot learning, demonstrating strong performance across various NLP tasks without fine-tuning. To measure the dynamic discrepancy between models, we apply KoopSTD on the hidden representation of their last layer, using the hyperparameters $l = 150, s = 1, r = 10$.

**LLaMA.** LLaMA (Large Language Model Meta AI) (Dubey et al., 2024) is a family of open-weight language models developed by Meta, designed for efficient and scalable natural language processing. It offers strong performance across various NLP tasks while being more accessible and resource-efficient than many proprietary models. In our experiments, we use LLaMA 3.2 for the 1B and 3B models and LLaMA 3.1 for the 8B model. The measured results by KoopSTD presented in Figure 6 can be reproduced by the hyperparameters $l = 50, s = 1, r = 10$.

## D. Detailed Experimental Results

**Structural functional relationship between cortical regions.** To better illustrate the functional-structural coupling patterns in the auditory cortex identified by KoopSTD in Sec. 4.4, we present the anatomical organization and myelin distribution of early auditory regions in Figure 11. This structural homogeneity aligns with their shared functional properties observed in our analysis, where A1, MBelt and LBelt areas are characterized by comparable high myelin content (shown in red-yellow). In contrast, RI and PBelt regions exhibit distinct myelination patterns, suggesting their relative independence in both structural and functional aspects. This anatomical evidence, particularly the myelin distribution pattern, provides additional support for our KoopSTD-based findings regarding the hierarchical organization and functional coupling within the auditory cortical network.

## E. Ablation Study

We demonstrate that KoopSTD better captures system dynamics compared to the recent HAVOK-based DSA (Ostrow et al., 2024), which relies on truncated-rank regression of time-delay embeddings (detailed in C.1). KoopSTD's advantage stems from two key aspects: decomposing system transitions across multiple timescales using time-frequency transformation, and filtering out spurious modes based on their spectral residuals. To validate, we conduct ablation studies on comparing different trajectories generated by the Lorenz system using HAVOK-based DSA, time-delay embedding with residual-controlled Koopman spectrum (an ablation), time-frequency transformation with truncated Koopman spectrum (another ablation), and the proposed KoopSTD.

Figure 12(a) demonstrates that both time-frequency transformation and residual-controlled Koopman spectrum contribute to KoopSTD's performance on the Lorenz system. By filtering spurious dynamical modes, KoopSTD successfully distinguishes

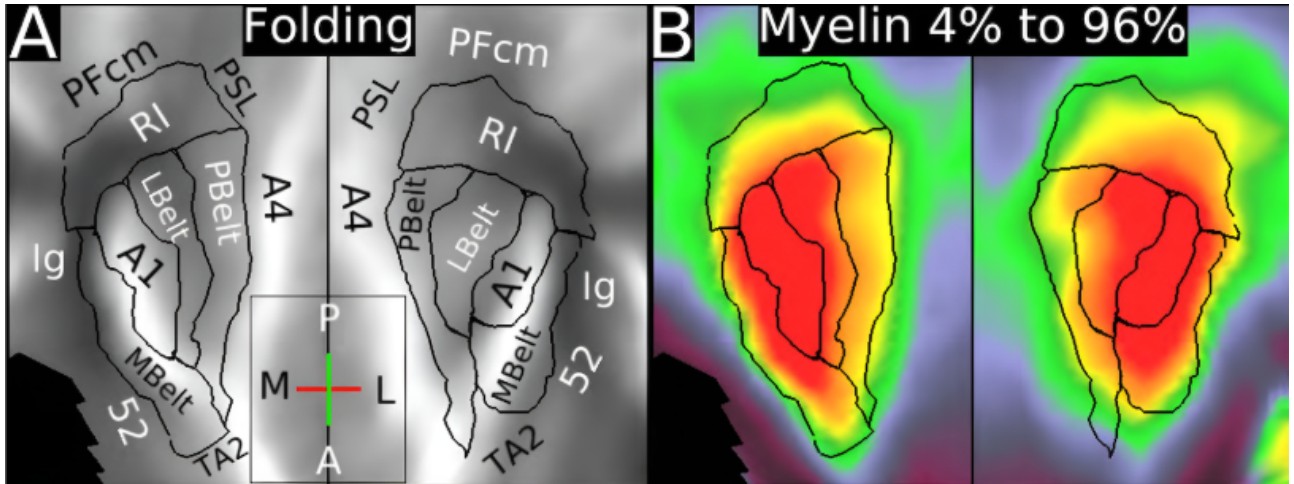

*Figure 11.* The early auditory regions are displayed on a folding map in Panel A, while Panel B presents myelin mapping scaled from 4% to 96% to emphasize variations in myelin density within the heavily myelinated auditory cortical regions (reproduced from (Glasser et al., 2016)).

between line-like and ring-like trajectories. This step results in a more concentrated eigenvalue distribution on the complex plane (*i.e.*, ring-like trajectory denoted by green triangles, Figure 12(b)), preserving only the modes that capture essential dynamics. Meanwhile, the time-frequency transformation effectively reveals the disparity between complex dynamical patterns through the multi-scale temporal analysis (*i.e.*, chaotic behavior denoted by red circles, periodic orbits denoted by yellow triangles and purple squares, Figure 12(c)).

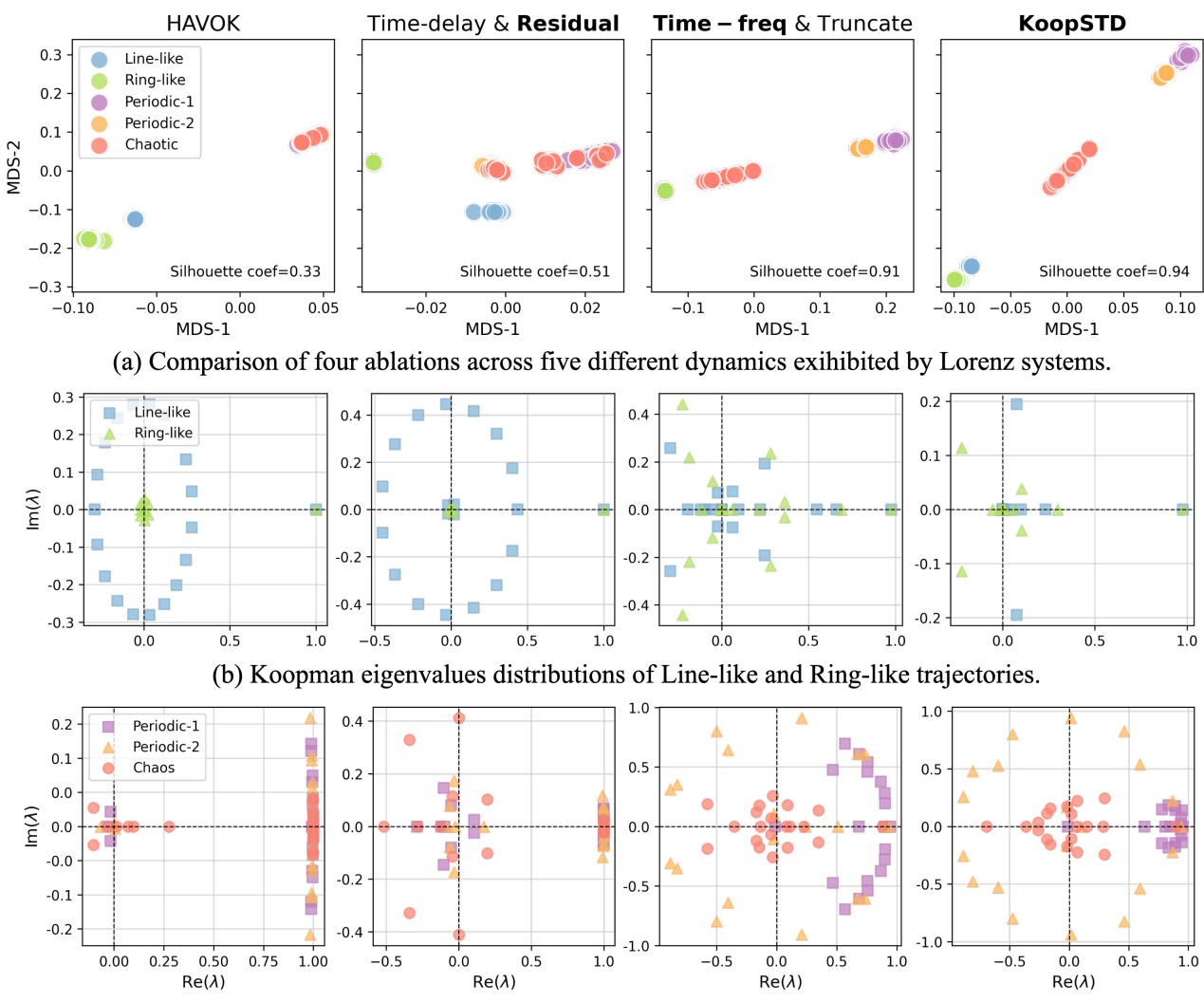

(a) Comparison of four ablations across five different dynamics exihibited by Lorenz systems.

(b) Koopman eigenvalues distributions of Line-like and Ring-like trajectories.

(c) Koopman eigenvalues distributions of Periodic and Chaotic trajectories.

*Figure 12.* Ablation study on time-frequency transformation and residual-controlled Koopman spectrum, each columns from left to right corresponds to one of the ablations.

