# OpenReview forum: "KoopSTD: Reliable Similarity Analysis between Dynamical Systems via Approximating Koopman Spectrum with Timescale Decoupling"
_ICML.cc/2025/Conference — ICML 2025 poster_

### Official Review · Reviewer_YjRQ · 2025-03-13

**Overall Recommendation:** 4

**Summary:**

This paper proposes KoopSTD, a new framework for comparing the similarity of non-linear dynamical systems by analyzing their Koopman spectra across multiple timescales. The main idea is to first transform time-series data into a time-frequency representation via short-time Fourier transforms, which extracts multi-scale dynamics. From there, the authors approximate the Koopman operator—a linear operator used to describe nonlinear system evolution—and apply a residual-based filtering step (Residual DMD) to remove “spurious” eigenvalues. The final step computes a similarity measure between the remaining, most accurate spectral components of any two dynamical systems. The paper demonstrates that KoopSTD (1) effectively separates different classes of Lorenz-type and other simulated chaotic dynamics, (2) robustly handles cases where geometric trajectories look similar but have different underlying dynamics (or vice versa), and (3) extends to real scenarios such as analyzing auditory cortex fMRI data and internal representations of large language models. Overall, the contributions center on using multi-scale Koopman analysis and spurious-mode filtering to yield a reliable, permutation-invariant metric of dynamical similarity.

**Claims And Evidence:**

Broadly, yes. The authors provide synthetic (Lorenz-like systems, attractors) and empirical (fMRI data, LLMs) evidence showing that KoopSTD better distinguishes or unifies multi-scale dynamical systems than existing metrics. They also demonstrate how Residual DMD helps filter spurious eigenvalues. While more large-scale or higher-dimensional results could further confirm generality, the current experiments convincingly support the paper’s main claims on multi-scale robustness and spectral reliability.

**Essential References Not Discussed:**

No

**Experimental Designs Or Analyses:**

I looked in detail at the setup of the Lorenz/attractor experiments, the 3-bit Flip-Flop RNN tasks, and the fMRI and LLM tests. Each uses a clear procedure: collecting time-series data, transforming with STFT, approximating the Koopman operator, filtering eigenvalues, then comparing metrics. The sampling sizes and time lengths appear sufficient to capture relevant dynamics, and the train/test splits or multiple runs (e.g., different initial seeds) help confirm robustness. Overall, the paper’s experiments are reasonably well-structured and consistent with their stated aims.

**Methods And Evaluation Criteria:**

Yes. The paper’s Koopman-based metric aligns naturally with multi-scale dynamical systems, and the authors employ Lorenz/attractor setups plus fMRI and LLM datasets, each demonstrating realistic temporal patterns. While relatively controlled, these examples effectively validate the method’s ability to handle different types of temporal complexity, which suits the paper’s goal of broadly characterizing non-linear dynamics across diverse settings.

**Other Comments Or Suggestions:**

NA

**Other Strengths And Weaknesses:**

NA

**Questions For Authors:**

NA

**Relation To Broader Scientific Literature:**

1. it builds on classical Koopman operator theory (Koopman, 1931; Kutz et al., 2016) and its practical approximation via dynamic mode decomposition (Williams et al., 2015), extending these ideas by incorporating short-time Fourier transforms to decouple dynamics across multiple timescales. This multi-scale analysis addresses the challenge of cross-scale coupling, which has been a known limitation in earlier Koopman-based metrics (e.g., Fujii et al., 2017; Ostrow et al., 2024).

2. By integrating Residual DMD techniques (Colbrook & Townsend, 2024), the paper tackles spectral pollution issues that have been highlighted in prior work (Lewin & Sére, 2010; Drmac et al., 2018).

3. This paper also relates to the broader literature on representation similarity analysis (e.g., Kornblith et al., 2019; Williams et al., 2021) by providing a robust, invariant measure for comparing the dynamics of complex systems such as neural and language models.

**Theoretical Claims:**

The proofs are correct. The proofs of the invariance of dissimilarity are detailed.

---

> ### Author Rebuttal · Authors · 2025-03-31
>
> We sincerely appreciate your thorough assessment of our work. We are delighted that you recognize KoopSTD's contributions to providing a robust and multi-scale similarity metric for nonlinear dynamical systems. In particular, we deeply appreciate your time and careful attention in examining our experimental setup, as well as the discussion of its theoretical foundations and empirical validation. Your recognition that our results convincingly support our claims is especially encouraging.
>
> Thank you again for your time and evaluation!

---

### Official Review · Reviewer_WLZa · 2025-03-13

**Overall Recommendation:** 4

**Summary:**

A method for computing dissimilarity between dynamical systems is proposed. It is based on the Wasserstein distance between sets of eigenvalues of the Koopman operators of the dynamics. Here two major heuristics are adopted. One is the STFT-based observable, which the authors motivate as a way to deal with multiple timescales of dynamics. The other is spectral residual as proposed in resDMD, with which the method focuses only on important eigenvalues and becomes more robust. The proposed method's utility is validated with different datasets.

### Update after rebuttal

Thank you for the detailed response. It would strengthen the paper to include the discussion about the phase information. I would keep my original positive rating.

**Claims And Evidence:**

The experiments nicely support the claim, i.e., ability to distinguish different dynamics and not to distinguish the same dynamics.

A bit tricky point seems to lie in the use of STFT. If I understood correctly, the method uses STFT's spectral magnitudes only and dispose the phase information (because in Algorithm 1, the space of $\mathcal{Z}$ is $\mathbb{R}$, not $\mathbb{C}$). The authors motivate the use of STFT with multiscale dynamics, but I think disposing the phase information is also an important feature of the method mainly for robustness. An ablation study dropping the STFT and instead taking time-dealy only is presented in appendix, which is nice, but I am curious about what happens if you do STFT but not dispose the phase, i.e., apply resDMD on the complex values outcome of STFT.

**Essential References Not Discussed:**

N/A

**Experimental Designs Or Analyses:**

(see Methods And Evaluation Criteria section)

**Methods And Evaluation Criteria:**

The method is simple yet reasonable. The empirical evaluation makes sense and is clearly presented.

**Other Comments Or Suggestions:**

N/A

**Other Strengths And Weaknesses:**

The proposed method is simple, which I think is a good feature as a method generally applicable in various domains. Probably the choice of the number of modes to use is somewhat tricky in practice, so some comments if any on it would be appreciated.

**Questions For Authors:**

N/A

**Relation To Broader Scientific Literature:**

Dynamics comparison appears in various field and is indeed an important problem. The paper should have decent impact in many fields.

**Theoretical Claims:**

The basic properties of the proposed dissimilarity are analyzed, which seem correct.

---

> ### Author Rebuttal · Authors · 2025-03-31
>
> We sincerely appreciate the reviewer's valuable feedback!
>
> Please kindly find the point-to-point response to the questions and suggestions.
>
> Q1: How about applying DMD on the complex part of STFT results instead of the real part?
>
> > We appreciate the reviewer’s insightful question. We agree that discarding phase information will enhance robustness, as phase is highly sensitive to time shifts and minor perturbations. This sensitivity is particularly problematic in fMRI and other biological signals, where recording conditions and preprocessing can introduce unavoidable phase shifts.
>
> > From the perspective of the Koopman operator theory, the magnitude, which represents the signal's energy distribution, generally evolves in a smoother manner and is more amenable to linear approximation, making it more suitable for DMD. On the other hand, phase information is often linked to transient and localized changes, which are more challenging to capture within the linear framework of the Koopman operator theory. Therefore, in the KoopSTD framework, the real part is more suitable.
>
> > We acknowledge that incorporating phase information in a controlled manner to enrich the description of system dynamics could be an interesting future direction. We will add a discussion on this point in the `Conclusion` section in the revised manuscript.
>
> > Once again, we sincerely thank the reviewer for bringing this to our attention.
>
> Q2: How can we determine the hyperparameters of KoopSTD in practice, including the number of modes?
>
> > Thanks for pointing out this question! We agree that the choice of hyperparameters — particularly the STFT window length $l$ and the number of preserved modes $r$ — deserves more discussion. We will add a brief description in the `Experimental Results` section in the revised main manuscript and a detailed discussion in the revised appendix.
>
> > In most DMD-related work, hyperparameters are typically chosen heuristically, relying on prior knowledge and experience with the specific system’s characteristics[1,2,3]. In contrast, by conducting an additional sensitivity analysis, we demonstrate that our KoopSTD remains robust across a wide range of these parameters, reducing the reliance on manual tuning. Specifically, we conducted the experiments on the classical Lorenz system, measuring the silhouette coefficients of the distance matrices obtained by KoopSTD with various $<l,r>$ combinations. In this setting, the silhouette coefficient$\in [-1,1]$ is the larger, the better.
>
> > The results are summarized in the table below. It is noteworthy that the results in most of the configurations are greater than 0.9. This indicates that KoopSTD is able to consistently distinguish different dynamical behaviors, even with diverse hyperparameter settings, highlighting its robustness in practice without requiring extensive hyperparameter tuning.
>
> | $l \, \backslash \, r$ |   5    |  10   |  15   |  20   |  25   |
> |------------------------|--------|-------|-------|-------|-------|
> | **500**                | 0.898  | 0.931 | 0.959 | 0.969 | 0.962 |
> | **600**                | 0.918  | 0.965 | 0.973 | 0.968 | 0.946 |
> | **700**                | 0.947  | 0.961 | 0.971 | 0.979 | 0.985 |
> | **800**                | 0.904  | 0.931 | 0.945 | 0.943 | 0.956 |
> | **900**                | 0.827  | 0.942 | 0.939 | 0.920 | 0.929 |
>
> > Note that increasing $r$ within a reasonable range preserves more meaningful modes, potentially enhancing the representation of the system's dynamic patterns. However, continuously increasing $r$ presents two key drawbacks:
> 1. Firstly, an excessively large number of modes can introduce uncontrollable spectral error, reducing the accuracy of the Koopman approximation.
> 2. Secondly, the computational cost can significantly rise. While the computational cost between a single pair comparison remains manageable, comparing all pairs in a large dataset becomes substantially more time-consuming.
>
> [1] Kutz et al., Multiresolution dynamic mode decomposition. In SIAM Journal on Applied Dynamical Systems, 2016.
>
> [2] Noack et al., Recursive dynamic mode decomposition of transient and post-transient wake flows. In Journal of Fluid Mechanics, 2016.
>
> [3] Brunto et al., Chaos as an intermittently forced linear system. In Nature communications, 2017.

---

### Official Review · Reviewer_ttZN · 2025-03-13

**Overall Recommendation:** 4

**Summary:**

This paper is part of the broad hot topic of comparing non-linear neural networks through the lens of representation space and focuses more specifically on the measurement of dissimilarity between non-linear dynamical systems.  These systems, due to their non-linearity, can be analyzed using the Koopman operator theory, which transforms them into a linear, infinite-dimensional space of observables. The authors introduce KoopSTD, a novel metric for assessing dynamical similarity by comparing their Koopman spectra. Their method involves 1) decomposing transitions in state space using the short-time Fourier transform (STFT), 2) extracting spatiotemporal patterns via Koopman mode decomposition (KMD), and 3) filtering out spurious modes by selecting the top $r$ eigenpairs of the Koopman operator with minimal spectral residuals.

The main contributions of the new metric are:

- Experiments: Applied to physical and neural systems, comparing it to existing similarity metrics like Centered kernel
alignment (CKA) (, Procrustes analysis, Cross-Correlation, Hankel
Alternative View of Koopman (HAVOK-based DSA).

- Theory: KoopSTD satisfies three invariance properties: isotropic scaling, rotation, and permutation invariance.

- Applications: Used for analyzing fMRI data and Large Language Models (LLMs).

**Claims And Evidence:**

The main claims regarding the new KoopSTD metric are as follows:

- Theoretical: The KoopSTD metric satisfies three key invariance properties: isotropic scaling (Theorem 1), rotation (Theorem 2), and permutation invariance (Theorem 3).

-  Experimental: The KoopSTD procedure outperforms existing similarity metrics (CKA, CC, Procrustes, HAVOK-DSA) in distinguishing five Lorenz systems with varying Rayleigh numbers $\rho$.  It is robust in capturing the underlying dynamical differences, even in the presence of similar or noisy attractors. Additionally, KoopSTD can correctly identify neural network systems with identical recurrent dynamics despite differences in their geometric network architectures.

- Prospects: The paper examines KoopSTD's ability 1) to uncover functional correspondences in the human auditory cortex and 2) to provide insights into the relationship between model size and the functional behavior of LLMs.


I believe the experimental section is quite convincing in demonstrating the effectiveness and capability of KoopSTD to distinguish (or identify) systems in situations where the dynamics show trajectory similarities (or differences in geometric architectures).

The work on using KoopSTD in computational neuroscience and LLMs, although exploratory, is interesting and could open up new perspectives.

However, the theoretical part seems less developed: the statements about invariance properties (isotropic scaling, rotation, and permutation), presented as 'theorems,' are on the level of textbook exercises, and their proofs need revision.

**Essential References Not Discussed:**

This comment is the same as the one in the previous section "Relation to broader scientific literature".

**Experimental Designs Or Analyses:**

I haven't verified the validity of the experimental designs in detail.

**Methods And Evaluation Criteria:**

The chosen benchmark datasets effectively showcase the procedure’s properties. Its effectiveness is demonstrated through the study of Lorenz systems with different Rayleigh numbers, leading to five distinct trajectory classes. KoopSTD’s ability to differentiate systems, even when their trajectories appear similar, is tested using three noisy attractors, where distinctions arise solely from noise-induced deviations. Additionally, experiments with three 3-bit Flip-Flop RNNs of varying architectures show that KoopSTD outperforms representation-based methods (CKA, Procrustes) and dynamical approaches (CC, HAVOK) in identifying neural network systems with the same recurrent dynamics, as measured by the Silhouette coefficient.

**Other Comments Or Suggestions:**

L161: I think it should be $g(x^t)=\sum_{i=1}^\infty\lambda_i^{\textcolor{red}{t}}\phi_i(x^0)v_i$.

L130 (second column): Frobenius.

L208: as $\mathbf{A_{tf}}=W_V'W_V^\dag$ (Equation (6)), it should be $\mathcal M=W_V^{'*}W_V'$, $\mathcal N=W_V^{*}W_V'$ and $\mathcal M=W_V^{*}W_V$.

L175(second column): to use eigenvalue\textbf{s}.

L250, Equation (10): The sequence of equalities is incorrect. It should be: $d(\mathcal C(\mathcal F_1,\mathcal F_2))=d(\mathcal F_1,\mathcal F_2)$.

 L250 (second column, table): PDM Attractors \textbf{Flip-Flop} RNNs.

L688: The Koopman matri\textbf{ces} $\mathbf A_1$ and $\mathbf A_2$.

 L705: The \textbf{system} $\mathcal F_1$ [...] transform\textbf{s} as follows.

 L776: I think it should be $\mathcal K$ instead of $K$.

 L790: Following Equation (30).

 L1120: five \textbf{different} dynamics.

**Other Strengths And Weaknesses:**

The main body is well-written and enjoyable to read. The application of KoopSTD to cortical fMRI data to highlight functional dissimilarities in auditory processing, aligning with myelination patterns, is particularly promising for its potential use in other areas.

The main weaknesses stated in other fields are the cause of my score. If the authors thoroughly revise the statements of the theorems and their proofs to address this question, I would be happy to reconsider my evaluation.

**Questions For Authors:**

The KoopSTD procedure is interesting and effective for studying similarity between dynamical systems, resistant to spurious eigenvalues, and has promising applications in neuroscience and LLMs. However, the presentation of the theoretical part is unconvincing in its current form. Is KoopSTD invariant to all similarity transformations (see section 'Theoretical claims')?

**Relation To Broader Scientific Literature:**

I believe there are recent works based on the related aera of \textit{learning Koopman operators} that provide precise spectral bounds,
Indeed, to effectively capture multi-scale temporal dynamics, the proposed KoopSTD first transforms the data into the time-frequency representation space via STFT, resulting in two time-frequency embeddings that extract the top $r$ eigenvalue-eigenvector pairs by minimizing spectral residuals (Equation (7)). The following papers provide bounds for the estimation errors of Koopman operators in different frameworks. I believe some of them (at least) should be cited; perhaps the estimation procedures developed in these works could allow for extracting the top $r$ eigenvalue-eigenvector pairs more efficiently than using a Galerkin approximation (which can be computationally expensive and can fail if the modes of the system are not well-separated in the frequency domain) followed by minimizing the residual error, as is done in the paper.

- Hou et al., Sparse learning of dynamical systems in RKHS: An operator-theoretic approach. In International Conference on Machine Learning, 2023. ($\beta$-mixing stochastic processes).
- Klus et al., Data-driven model reduction and transfer operator approximation Journal of Nonlinear Science, 2018. (Time-homogeneous stochastic processes).
- Klus et al., Eigendecompositions of transfer operators in reproducing kernel Hilbert
spaces. Journal of Nonlinear Science, 30(1):283–315,
2020c
- Kostic et al., Learning dynamical systems via Koopman operator regression in reproducing kernel Hilbert spaces, NeurIPS, 2022. (Markov chains)
- Kostic et al., Sharp spectral rates for Koopman operator learning, NeurIPS, 2023. (Time-reversal-invariant stochastic dynamical systems).
- Philipp et al., Error bounds for kernel-based approximations of the Koopman operator}. Applied and Computational Harmonic Analysis, 2024. (Stochastic differential equations).
- Bevanda et al. Koopman Kernel Regression In Advances in Neural Information Processing Systems 36 (deterministic dynamical systems)
- Drmac and Mezic, A data driven Koopman-Schur decomposition for computational analysis of nonlinear dynamics, arXiv preprint arXiv:2312.15837

**Theoretical Claims:**

There are several issues with the theoretical claims and their proofs. The concept of __Koopman matrix__  is used in improper manner. In author's notation, for system $\mathcal F_1$ the Koopman matrix $A_1$ is an __approximation of an operator__ and equation  $X_{t+1} = \mathbf A_1 X_t$ makes no sense at all. The systems are not linear autonomous systems (which is the meaning of the equation). The __embeddings/representations  are crucial part__ of Koopman based approaches, and there they are not treated properly.

Assuming one correctly writes whole operator formalism, and assuming that one proves the metric induced by the operators and not approximations (which is not in my opinion sufficient to imply guarantee theoretical guarantees of the method, but only suggest its groundedness), still theorems would need rewriting. Namely:

__Theorem 1__: The equality $d(c\lambda(\mathbf A_1),c\lambda(\mathbf A_2))=d(\lambda(\mathbf A_1),\lambda(\mathbf A_2))$ seams to be false. Using it, it follows immediately from definition (8) that $d(c\mathcal F_1,c\mathcal F_2)=|c|d(\mathcal F_1,\mathcal F_2)$. Moreover, the notation $d(c\mathcal F_1,c\mathcal F_2)$ is clumsy and the proof is unclear as it stands. Similarly, Equations (17) (proof of Theorem 2) and (18) (proof of Theorem 3) should be rewritten in a clearer and more understandable way.


__Theorems 1-3__: All results are in fact a direct consequence of the well-known fact that the characteristic polynomial is invariant under similarity transformations. In particular, similar matrices have the same eigenvalues and, therefore, the same characteristic polynomial.Similarity transformations include scaling (Theorem 1: $A_1^c=DAD^{-1}$ where $D=cI$), rotation (Theorem 2), \textit{shearing} (if $\mathbf {A}_1^{\text{sheared}}=P\mathbf {A}_1P^{-1}$ for some invertible matrix $P$), and orthogonal transformations (Theorem 3).

---

> ### Author Rebuttal · Authors · 2025-03-31
>
> Thanks for your careful review and comments!
>
> Glad to see that you deem our work interesting and have a promising impact on many fields. Please kindly find our point-to-point responses below.
>
> ## Concerns
> Q1: Presenting the invariance property as 'theorem'.
>
> >We revise the invariance property from "Theorem” to "Proposition”.
>
> Q2: Using "theoretical guarantees" to describe the properties of KoopSTD.
>
> > We revise the "theoretical guarantees" to "theoretical groundedness" accordingly.
>
> Q3: Improper usage of "Koopman matrix" and "$𝐗_{t+1}=𝐀_{1}𝐗_{t}$".
>
> > We correct the terminology to "approximated Koopman operator" and "$g(𝐗_{t+1})=𝐀_{1}(g(𝐗_{t}))$", respectively.
>
> Q4: Incorrect equality $d(cλ(𝐀_1),cλ(𝐀_2))=d(λ(𝐀_1),λ(𝐀_2))$.
>
> > We fix this error by removing line 250 (left) in the main manuscript to let $d(c𝓕_1,c𝓕_2)=d(λ(𝐀_1),λ(𝐀_2))$ directly.
>
> Q5: Unclear proof of three invariance properties.
>
> > We refine our proof by unifying three invariance properties into a single proposition below.
>
> **Proposition.** Let $𝐗_1[t+1]= 𝓕_1({𝐗_1[t]})$ and $𝐗_2[t+1]= 𝓕_2({𝐗_2[t]})$ be two time-discrete dynamical systems, where $𝓕_1, 𝓕_2 : ℝ^{N_d} \to ℝ^{N_d}$. The dissimilarity $d(𝓕_1, 𝓕_2)$, as measured by KoopSTD, remains invariant under invertible linear transformations 𝓣,
> $$
>     d(𝓣(𝓕_1, 𝓕_2)) = d(𝓕_1, 𝓕_2),
> $$
> where $𝓣=${$𝐗 ↦ 𝐗𝐐: 𝐐 \in GL(N_d, ℝ)$}.
>
> **Proof.** For a dynamical system $𝐗[t+1]= 𝓕({𝐗[t]})$. The KoopSTD seeks to obtain the approximate Koopman operator $𝐀$ by firstly mapping $𝐗\inℝ^{T\times N_d}$ into a time-frequency representation $𝐙\inℝ^{(\frac{T-l}{s}+1)\times(\frac{l}{2}+1)\times N_d}$ using STFT.
>
> By the linearity of the STFT, we can express the process of STFT as:
> $$
>     𝐙=\text{STFT}(𝐗)=𝓛_{tf}𝐗,
> $$
> where $𝓛_{tf}$ refers to the linear operator of the STFT.
>
> Then, KoopSTD computes the right singular vectors $𝑽$ of $𝐙$ by solving SVD:
> $$
>    𝐙^* 𝐙=(𝑽\Sigma 𝑼^*)(𝑼\Sigma 𝑽^*)=𝑽\Sigma^2𝑽^*.
> $$
>
> Therefore, the Koopman operator can be approximated based on temporal snapshots $𝐖_{𝑽}$ of $𝑽$:
> $$
>     𝐀 = 𝐖_𝑽'𝐖_𝑽^{†}.
> $$
>
> Given any transformed input $𝐗𝐐$, we can also derive its time-frequency representation $𝐙_{𝓣}$:
> $$
>     𝐙_{𝓣}=\text{STFT}(𝐗𝐐)=𝓛_{tf}𝐗𝐐=𝐙𝐐.
> $$
>
> Then, the transformed singular vectors $𝑽_{𝓣}$ of $𝐙_{𝓣}$ can be derived as follows:
> $$
>     𝐙_{𝓣}^* 𝐙_{𝓣}=(𝐙𝐐)^* 𝐙𝐐=𝐐^* 𝐙^* 𝐙𝐐 =(𝐐^* 𝑽)\Sigma^2(𝑽^* 𝐐),
> $$
> $$
>     𝑽_{𝓣}=𝐐^*𝑽.
> $$
>
> The transformed snapshots $𝐖_{𝑽,𝓣}$ of $𝑽_{𝓣}$ can also be derived:
> $$
>     𝐖_{𝑽,𝓣}=𝐐^*𝐖_{𝑽},
> $$
>
> $$
>     𝐖'_{𝑽,𝓣}=𝐐^*𝐖'_𝑽.
> $$
>
> Thus, the transformed approximated Koopman operator $𝐀_{𝓣}$ can be denoted as follows:
> $$
>     𝐀_{𝓣}=𝐖_{𝑽,𝓣}'𝐖_{𝑽,𝓣}^{†}=𝐐^* 𝐀(𝐐^*)^{-1}.
> $$
>
> By definition of similarity transformation, the matrices $𝐀_{𝓣}$ and $𝐀$ are similar, implying that they share the same eigenvalues:
> $$
>     λ(𝐀_{𝓣})=λ(𝐀).
> $$
> Based on Definition (1) and the above derivation, we have:
> $$
>     d(𝓣(𝓕_{1},𝓕_{2}))= d(λ(𝐀_{1,𝓣}),λ(𝐀_{2,𝓣}))=d(λ(𝐀_{1}), λ(𝐀_{2})) = d(𝓕_1,𝓕_2).
> $$
> Since the following transformations on the feature dimension are also invertible linear transformations, we can deduce that KoopSTD remains invariant to the following transformations,
> - **Isotropic scaling**$:𝓣_{IS}=${$𝐗↦ 𝐗𝐐:𝐐=qI_{N_d},q\in ℝ^+$}.
> - **Rotations**$:𝓣_{R}=${$𝐗 ↦ 𝐗𝐐:𝐐 \in O(N_d)$}, where $O(N_d) :=$ {$ℝ^{N_d \times N_d}, 𝐐^T𝐐=I_{N_d}$} denote the orthogonal group.
> - **Permutations**$:𝓣_{P}=${$𝐗 ↦ 𝐗𝐐_{\pi}: \pi \in P(N_d)$}, where $P(N_d)$ is the set of permutations on {$1,...,N_d$}.
>
> Q6: The properties of KoopSTD stem from the invariance of the characteristic polynomial under similarity transformations. Does KoopSTD remain invariant to all similarity transformations?
>
> > As shown in our response to Q5, applying any invertible linear transformation to $𝐗$ is equivalent to applying a similarity transformation on the approximated Koopman operator $𝐀$, thereby ensuring KoopSTD's measurement remains invariant to all invertible linear transformations.
>
> > We emphasized the invariance properties in terms of *isotropic scaling, rotations and permutations* in the manuscript because they are essential for developing a reliable metric that ensures **robust** similarity measurements across different systems.
>
> ## Related work of spectral error estimation provided by the reviewer
>
> > Thank you for providing insightful related work!
>
> > We agree that the performance of spectral error estimation using the Galerkin method can be influenced by the choice of basis. We have carefully read these relevant studies. The `Introduction` section will cover them briefly. Additionally, we will cite some of them in the `Conclusion` section as future work in effectively extracting the reliable modes.
>
>
> ## Typos pointed out by the reviewer
> > Thanks for your careful review! We have proofread the manuscript, the typos will be fixed accordingly.

---

> > ### Comment · Reviewer_ttZN · 2025-04-03
> >
> > I thank the authors for their reply. I rise my score to Accept.

---

> > > ### Author Response · Authors · 2025-04-04
> > >
> > > Thank you for your prompt feedback! We sincerely appreciate your constructive comments, which have helped improve our work.

---

### Decision · Program_Chairs · 2025-05-01

**Decision:**

Accept (poster)

**Comment:**

This paper introduces KoopSTD, a novel method for comparing nonlinear dynamical systems by approximating their Koopman spectra with a timescale-decoupling approach. The method leverages STFT-based observables to address cross-scale coupling and applies residual-based filtering to isolate meaningful spectral components. The paper demonstrates the utility of KoopSTD through well-structured experiments on physical systems, neural networks, fMRI data, and large language models, showing it to be both effective and robust across settings. While the theoretical exposition—particularly regarding invariance properties—was initially underdeveloped, the authors responded thoroughly by refining proofs, clarifying terminology, and addressing reviewer concerns in detail. The method’s simplicity, general applicability, and potential for impact across neuroscience, machine learning, and system analysis make it a valuable contribution. I recommend acceptance.